# *Trans* regulation of an odorant binding protein by a proto-Y chromosome affects male courtship in house fly

Pablo J Delclos*†‡, Kiran Adhikari†§, Alexander B Mai, Oluwatomi Hassan, Alexander A Oderhowho#, Vyshnika Sriskantharajah¶, Tammie Trinh**, Richard Meisel

Department of Biology & Biochemistry, University of Houston, Houston, United States

**\*For correspondence:**
delclosp@uhd.edu

†These authors contributed equally to this work

**Present address:** ‡Department of Natural Sciences, University of Houston-Downtown, Houston, United States; §Oxford Biomedica Solutions LLC, Bedford, United States; #Department of Neurosurgery, Ochsner LSU Health Shreveport, Shreveport, United States; ¶University of Texas Health Science Center School of Biomedical Informatics, Houston, United States; **University of Texas School of Dentistry, Houston, United States

**Abstract** The male-limited inheritance of Y chromosomes favors alleles that increase male fitness, often at the expense of female fitness. Determining the mechanisms underlying these sexually antagonistic effects is challenging because it can require studying Y-linked alleles while they still segregate as polymorphisms. We used a Y chromosome polymorphism in the house fly, *Musca domestica*, to address this challenge. Two male determining Y chromosomes (Y^M and III^M) segregate as stable polymorphisms in natural populations, and they differentially affect multiple traits, including male courtship performance. We identified differentially expressed genes encoding odorant binding proteins (in the *Obp56h* family) as candidate agents for the courtship differences. Through network analysis and allele-specific expression measurements, we identified multiple genes on the house fly III^M chromosome that could serve as *trans* regulators of *Obp56h* gene expression. One of those genes is homologous to *Drosophila melanogaster CG2120*, which encodes a transcription factor that binds near *Obp56h*. Upregulation of *CG2120* in *D. melanogaster* nervous tissues reduces copulation latency, consistent with this transcription factor acting as a negative regulator of *Obp56h* expression. The transcription factor gene, which we name *speed date*, demonstrates a molecular mechanism by which a Y-linked gene can evolve male-beneficial effects.

## Editor's evaluation

This valuable study investigated the effect of different levels of expression of odorant proteins from the Obp56h family on behaviour depending on the locus encoding the protein, either on the sex proto-chromosomes III or Y of the house fly. The set of evidence obtained by combining observations in house fly and functional test in the model fly *Drosophila melanogaster* is solid, but the causal effect of the odorant protein expression and the fly behaviour will need to be functionally validated in house fly as well. With the causal link established directly, this paper would be of outstanding interest to evolutionary biologists and geneticists working on the dynamics of sexual selection.

## Introduction

In species with genetic sex determination, a Y or W chromosome can have sex-limited inheritance (*Bachtrog et al., 2014*; *Beukeboom and Perrin, 2014*). The male-limited inheritance of Y chromosomes is predicted to allow for the accumulation of alleles with male-specific beneficial effects (*Rice, 1996*). These male-beneficial alleles can have female-deleterious (sexually antagonistic) effects because they are never exposed to direct selection in females (*Charlesworth et al., 2005*; *Abbott et al., 2017*).

One important way that sex-limited Y and W chromosomes appear to affect sex-specific traits is via *trans* regulation of genes elsewhere in the genome. For example, Y chromosome genotypes in *Drosophila melanogaster* have *trans* effects on gene expression throughout the genome, which modify a broad range of phenotypes, including immunity and chromatin (*Lemos et al., 2008*; *Lemos et al., 2010*; *Brown et al., 2020*). *D. melanogaster* Y chromosome genotypes also have fitness consequences that depend on the genetic background, suggesting epistatic interactions between Y-linked alleles and the X or autosomes (*Chippindale and Rice, 2001*). Similarly, in *Poecilia* spp., Y-linked alleles may affect sexually selected male pigmentation patterns by acting as *trans* regulators of autosomal gene expression (*Morris et al., 2020*; *Kawamoto et al., 2021*; *Sandkam et al., 2021*), although the specific mechanisms of these effects are not well understood.

Even though we are aware of sex-specific phenotypic and fitness effects of Y and W chromosomes, the mechanisms underlying these effects are not as well understood. Notably, we have a limited understanding of how Y and W chromosomes act as *trans* regulators of sex-specific and sexually antagonistic traits genome-wide. There are also very few sexually antagonistic alleles that have been genetically or molecular characterized on young sex chromosomes (cf. *Roberts et al., 2009*), which limits our ability to make generalizations about the molecular mechanisms underlying sexually antagonistic selection on Y and W chromosomes (*Mank et al., 2014*). To address these shortcomings, we sought to identify the genetic mechanism underlying how a young Y chromosome affects male courtship behavior in the house fly, *Musca domestica*.

House fly has a multifactorial sex determination system, in which multiple young proto-Y and proto-W chromosomes segregate as polymorphisms in natural populations (*Hamm et al., 2014*). The two most common proto-Y chromosomes (III$^M$ and Y$^M$) are distributed along latitudinal clines on multiple continents (*Franco et al., 1982*; *Tomita and Wada, 1989*; *Hamm et al., 2005*), and they affect thermal traits in ways that are consistent with their geographic distributions (*Delclos et al., 2021*). This polymorphism has remained stable in natural populations since at least the mid-20th century, suggesting that selection maintains multifactorial sex determination (*Kozielska et al., 2008*; *Meisel et al., 2016*).

Here, we focus on the effect of the house fly proto-Y chromosomes on male courtship performance. Males carrying the III$^M$ chromosome (III$^M$ males) outcompete males carrying the Y$^M$ chromosome (Y$^M$ males) for female mates (*Hamm et al., 2009*). In light of this and other aforementioned phenotypic differences between Y$^M$ and III$^M$ males, it is remarkable that the III$^M$ and Y$^M$ chromosomes carry nearly all of the same genes as their homologous proto-X chromosomes (III and X), and only a small number of allelic differences have been identified (*Meisel et al., 2017*; *Son and Meisel, 2021*). These similarities between the proto-Y and proto-X chromosomes have led to the hypothesis that the phenotypic effects of the proto-Y chromosomes may be mediated by *trans* effects of Y$^M$ and III$^M$ alleles on the expression of genes elsewhere in the genome (*Adhikari et al., 2021*). We aimed to identify the *trans* regulatory allele(s) on the Y$^M$ or III$^M$ chromosomes that affect differences in courtship performance between the genotypes.

## Results

### Differential expression of odorant binding protein genes between III$^M$ and Y$^M$ males

Our first goal was to analyze RNA-seq data in order to identify differentially expressed (DE) genes between III$^M$ and Y$^M$ male heads that could be responsible for the previously observed differences in competitive courtship assays between the genotypes (*Hamm et al., 2009*). We confirmed that the gene expression profiles of III$^M$ and Y$^M$ male heads are minimally differentiated (*Meisel et al., 2015*; *Son et al., 2019*). There are only 40 DE genes between heads of III$^M$ and Y$^M$ adult males (21 upregulated in Y$^M$ males, 19 upregulated in III$^M$, *Supplementary file 1C*). Gene ontology (GO) analysis revealed no significant biological process, molecular function, or cellular component terms enriched within the list of DE genes.

Within the list of DE genes, we identified one gene (*LOC105261916*) encoding an odorant binding protein (Obp) upregulated in Y$^M$ males. House fly Obp genes can be grouped into families corresponding to their *D. melanogaster* orthologs (*Scott et al., 2014*). The DE Obp gene in our analysis is orthologous to *Obp56h*. The *Obp56h* family, as well as other Obp families, was greatly expanded

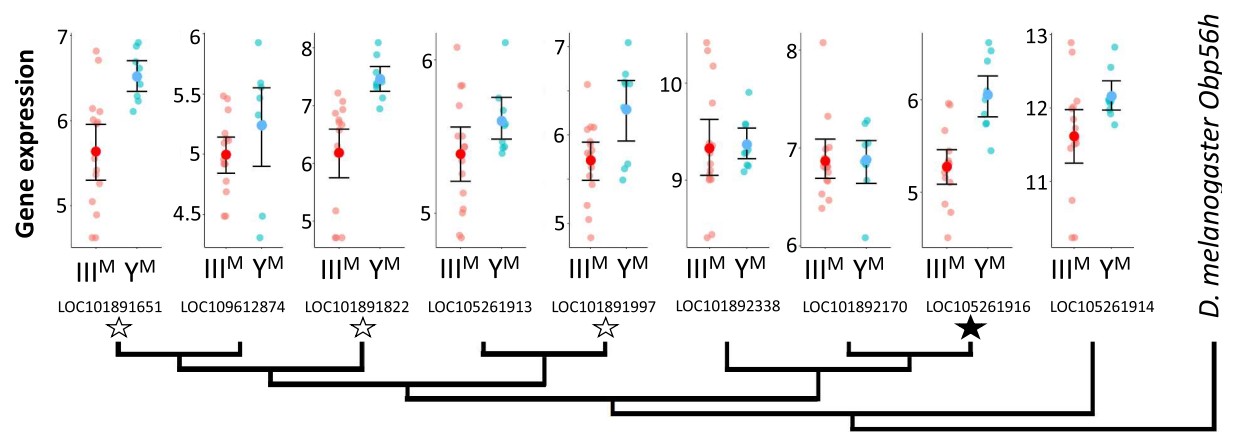

**Figure 1.** Neighbor-joining phylogenetic tree of the *Obp56h* gene family within *M. domestica* and *D. melanogaster* based on protein sequences constructed in MEGA X (*Kumar et al., 2018*). Amino acid sequences were aligned by MUSCLE (*Edgar, 2004*). *M. domestica Obp56h* genes are identified based on gene IDs. The bootstrap consensus tree was inferred from 10,000 replicates. Branch lengths are scaled according to the number of amino acid substitutions per site. The phylogeny was arbitrarily rooted at *D. melanogaster Obp56h*. Graphs at the branch tips show batch-adjusted expression levels for each *M. domestica Obp56h* gene from each replicate (small circles). Large circles show the average across all replicates, with error bars denoting the standard error (unfilled stars: p<0.05 before false discovery rate [FDR] correction for multiple comparisons; filled star: p<0.05 after correction).

The online version of this article includes the following figure supplement(s) for figure 1:

**Figure supplement 1.** Summary of *Obp56h* expression in III$^M$ (red) and Y$^M$ (blue) males reared at 18°C and 29°C.

within Muscidae (house fly and close relatives, including stable fly and horn fly) compared to *D. melanogaster* (*Scott et al., 2014*; *Olafson and Saski, 2020*; *Olafson et al., 2021*). In addition to *LOC105261916*, seven of the remaining eight house fly *Obp56h* genes for which we obtained RNA-seq count data showed similar trends of greater expression in Y$^M$ than III$^M$ males, with three of these showing significant DE (p<0.05) before a false discovery rate (FDR) correction (*Figure 1*). All but one of the *Obp56h* genes has higher expression in Y$^M$ than III$^M$ males (8/9, regardless of significance), which is significantly greater than the fraction of other genes with higher expression in Y$^M$ males, regardless of significance, in the rest of the genome (Fisher's exact test, p=0.019).

## III$^M$ confers a courtship advantage by reducing copulation latency

Knockdown of *Obp56h* in *D. melanogaster* decreases male copulation latency, or the time it takes for a male to begin to mate with a female after they are first exposed to one another (*Shorter et al., 2016*). The *Obp56h* gene family is generally expressed higher in Y$^M$ males relative to III$^M$ males (*Figure 1*). A previous study identified a competitive advantage of III$^M$ over Y$^M$ male house flies in successfully engaging a female in mating (*Hamm et al., 2009*), consistent with shorter copulation latency in III$^M$ males because of lower expression of *Obp56h* genes. We confirmed this result by performing competitive courtship assays in which we allowed males carrying III$^M$ or Y$^M$ to compete for a female of an unrelated strain. Consistent with the previous results, we observed that III$^M$ males are more successful at mating than Y$^M$ males when raised at 29°C (*Figure 2A*), a temperature similar to the developmental temperature used by *Hamm et al., 2009*.

Two *Obp56h* genes are only upregulated in Y$^M$ males at 29°C, but not at 18°C (*Figure 1—figure supplement 1*), raising the possibility that the effect of III$^M$/Y$^M$ genotype on male courtship success may be temperature-dependent. In addition, III$^M$ males have greater heat tolerance than Y$^M$ males (*Delclos et al., 2021*), further suggesting that the benefits of the III$^M$ chromosome may be limited to warm temperatures. We therefore tested if the differences in courtship performance between Y$^M$ and III$^M$ males are sensitive to temperature. We found that III$^M$ males were more successful at mating than Y$^M$ males regardless of developmental temperature (ANOVA, $\chi^2$=20.7, p=6.53 × 10$^{-6}$, *Figure 2A*). We then tested whether males reared at different developmental temperatures, but with the same genotype, have a difference in courtship success. We found that males reared at 18°C outcompete males reared at 29°C, regardless of genotype (ANOVA, $\chi^2$=13.1, p=2.93 × 10$^{-4}$, *Figure 2B*). This

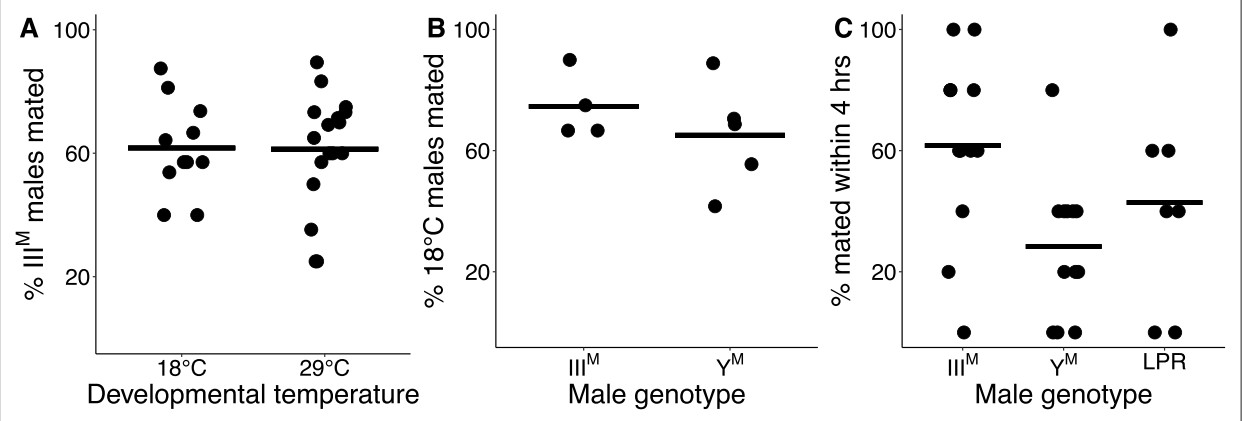

**Figure 2.** III^M chromosome and developmental temperature affect male courtship success. (**A**) Outcomes of competitive courtship assays between III^M and Y^M males reared at 18°C or 29°C. Data points represent experimental batches. Horizontal lines denote the median across all batches. (**B**) Outcomes of competitive courtship assays conducted between males reared at 29°C and 18°C. Trials were conducted between males of the same proto-Y chromosome genotype (III^M or Y^M). Each data point represents 10 replicate trials within a single batch. (**C**) Outcomes of single-choice courtship assays in males reared at 22°C. Data points refer to the percentage of males (five males within one replicate) that mated with females within 4 hr within each experimental trial. Horizontal lines denote means within male groups. All females used were from the LPR strain.

The online version of this article includes the following figure supplement(s) for figure 2:

**Figure supplement 1.** Summary of the effect of developmental temperature on copulation latency in III^M (salmon), Y^M (turquoise), and LPR (black) males reared at 22°C and 29°C.

**Figure supplement 2.** Summary of survival analysis results between male proto-Y chromosome genotypes and developmental temperatures.

demonstrates that there is an effect of developmental temperature on courtship success, but it appears to be independent of the genotype effect. More generally, our results suggest that the III^M male courtship advantage is robust to developmental temperature, which differs from the prediction based on temperature-dependent *Obp56h* expression (*Figure 1—figure supplement 1*).

We next tested if the effects of developmental temperature and genotype on competitive mating advantage could be caused by differences in copulation latency. To measure copulation latency, we combined five males from a single strain raised at a single temperature with five females from the unrelated strain used in our competitive courtship assays. Developmental temperature had a significant effect on copulation latency (ANOVA, $\chi^2$=15.3, p=9.40 × 10^{-5}), with males reared at 22°C mating faster than those reared at 29°C (*Figure 2—figure supplement 1A*). This result is consistent with increased courtship success of males raised at 18°C relative to those raised at 29°C in our competitive assays (*Figure 2B*).

We conducted a survival analysis to test for differences in copulation latency between Y^M and III^M. To those ends, we fit a Cox proportional hazards regression model, treating unmated males as censored data (*Burke and Holwell, 2021*), and we observed significant effects of both developmental temperature (ANOVA, $\chi^2$=14.6, p=1.35 × 10^{-4}) and male genotype ($\chi^2$=15.8, p=3.73 × 10^{-4}) on copulation latency. However, there was no significant interaction effect between genotype and developmental temperature ($\chi^2$=3.86, p=0.145). Consistent with our hypothesis that III^M males have a reduced copulation latency relative to Y^M males, at 22°C, III^M males mated nearly three times faster than Y^M males (Cox model: HR = 2.92, 95% CI=1.42–6.03, *Figure 2—figure supplement 2A*). We do not observe this trend at 29°C (*Figure 2—figure supplement 2B*), possibly due to very few successful matings at the warmer developmental temperature.

To further address the limitations of censored data, we treated copulation latency as a binary variable by calculating the proportion of the five males per trial that mated within the 4 hr assay. We observed significant effects of male genotype (ANOVA, $\chi^2$=10.2, p=6.18 × 10^{-3}) and developmental temperature (ANOVA, $\chi^2$=11.0, p=9.04 × 10^{-4}) on the proportion of males that mated. The effect of developmental temperature was largely a result of very few matings for males that developed at 29°C relative to 22°C (*Figure 2—figure supplement 1B*), consistent with our finding that males that developed at a lower temperature have higher courtship success (*Figure 2B*). In the 22°C treatment, a significantly greater proportion of III^M males mated within 4 hr than Y^M (61.7% vs. 28.3%; Z-test of

proportions, $p=1.21 \times 10^{-4}$; *Figure 2C*). There is further evidence that III$^M$ males have reduced copulation latency, which is consistent with their previously documented competitive advantage (*Hamm et al., 2009*), the competitive advantage that we observe (*Figure 2A*), and the reduced expression of *Obp56h* genes (*Figure 1*).

The Y$^M$ and III$^M$ males that we and others have previously used in courtship experiments all share the CS genetic background that comes from a III$^M$ strain (*Hamm et al., 2009*). This raises the possibility that III$^M$ males perform better because they have a proto-Y chromosome that is co-adapted to its genetic background. To test this hypothesis, we measured copulation latency in Y$^M$ males from the same strain (LPR) as the females in our experiments. We observed a greater proportion of III$^M$ males mating within 4 hr when compared to the LPR Y$^M$ males (Z-test of proportions, $p=0.038$), although the copulation latency in LPR males was highly variable (*Figure 2C*). Therefore, the reduced copulation latency conferred by the III$^M$ chromosome overwhelms any potential effects of co-adaptation of the proto-Y chromosome to male genetic background or male-female co-adaptation within strains. The reduced copulation latency of III$^M$ males is only detectable when house flies develop at 22°C, suggesting that it is either temperature-dependent or we lack the resolution to detect it when males develop at warmer temperatures (because they take too long to mate).

## House fly chromosome III genes and *Drosophila* X chromosome genes have correlated expression with Obp56h genes

The *Obp56h* gene cluster is found on house fly chromosome V, suggesting that it is regulated in *trans* by genes on the Y$^M$ or III$^M$ chromosome. Chromosome V is unlikely to differ between the Y$^M$ and III$^M$ males in our experiments: although not completely genetically identical, the majority of males compared in the RNA-seq data and mating experiments have a common genetic background (including chromosome V) and differ primarily in whether they carry III$^M$ or Y$^M$. Removing samples with a different genetic background did not affect the general difference in *Obp56h* expression between III$^M$ and Y$^M$ males. Specifically, we conducted DE analysis on males with a shared (CS) genetic background. These males with a shared genetic background differ only in their proto-Y chromosome (Y$^M$ or III$^M$). In this subset of our full data set, we identified 12 significantly DE genes between III$^M$ and Y$^M$ males, all of which are also DE in the full data set. The same *Obp56h* gene (*LOC105261916*) that was DE after Benjamini-Hochberg p-value adjustment in the full data set was also DE here. In addition, five more *Obp56h* genes had raw p-values<0.05 in this data set, compared to 3 with raw p<0.05 in the full data set. Due to low power from the smaller number of RNA-seq samples, we did not perform weighted gene co-expression network analysis (WGCNA) in this truncated data set. We therefore focused on the full data set to identify genes on the Y$^M$ or III$^M$ chromosome that could be responsible for the differential expression of *Obp56h* genes between Y$^M$ and III$^M$ males.

We first identified 27 co-expression modules across Y$^M$ and III$^M$ house fly male heads. We focus on one of these modules (containing 122 genes, *Supplementary file 1D*) because it is DE between Y$^M$ and III$^M$ males ($p_{ADJ} = 0.001$), and it contains three *Obp56h* genes that are DE between III$^M$ and Y$^M$ males (*LOC105261916, LOC101891822,* and *LOC101891651*) (*Figure 3*). GO analysis revealed significant enrichment ($p_{ADJ}<0.05$) of 15 biological process terms including those related to immune system processes (GO:0032501), responses to stress (GO:0006950), and response to external stimuli (GO:0009605) within this module (*Supplementary file 1E*). We used the WGCNA measure of intramodular connectivity, kWithin, to identify hub genes within the module that likely have important roles in the regulation of gene expression. The top five hub genes are (with *D. melanogaster* orthologs in parentheses): *LOC101887703* (*CG8745*), *LOC105262120* (*CG10514*), *LOC101894501* (*Md-Gr35*), *LOC101893264* (*gd*), and *LOC101893651* (*CG2120*) (*Figure 3—figure supplement 1*).

The genes present in the focal co-expression module provide multiple lines of evidence that *Obp56h* gene expression is regulated by *trans* factors that map to chromosome III. First, the module is enriched for house fly chromosome III genes (31 chromosome III genes vs. 38 genes assigned to other chromosomes, Fisher's exact test $p<1 \times 10^{-5}$, with 53 genes not assigned to a chromosome) and for DE genes between Y$^M$ and III$^M$ males (16 DE genes in this module vs. 24 DE genes assigned to other modules, Fisher's exact test $p<1 \times 10^{-5}$). In addition, the co-expression module is enriched for *Obp56h* genes relative to other Obp genes—three *Obp56h* genes and no other Obp genes were assigned to this module (Fisher's exact test, $p=5.1 \times 10^{-3}$, *Figure 3*). Furthermore, there is a significant enrichment of chromosome III genes within the 100 genes whose expression covaries most with

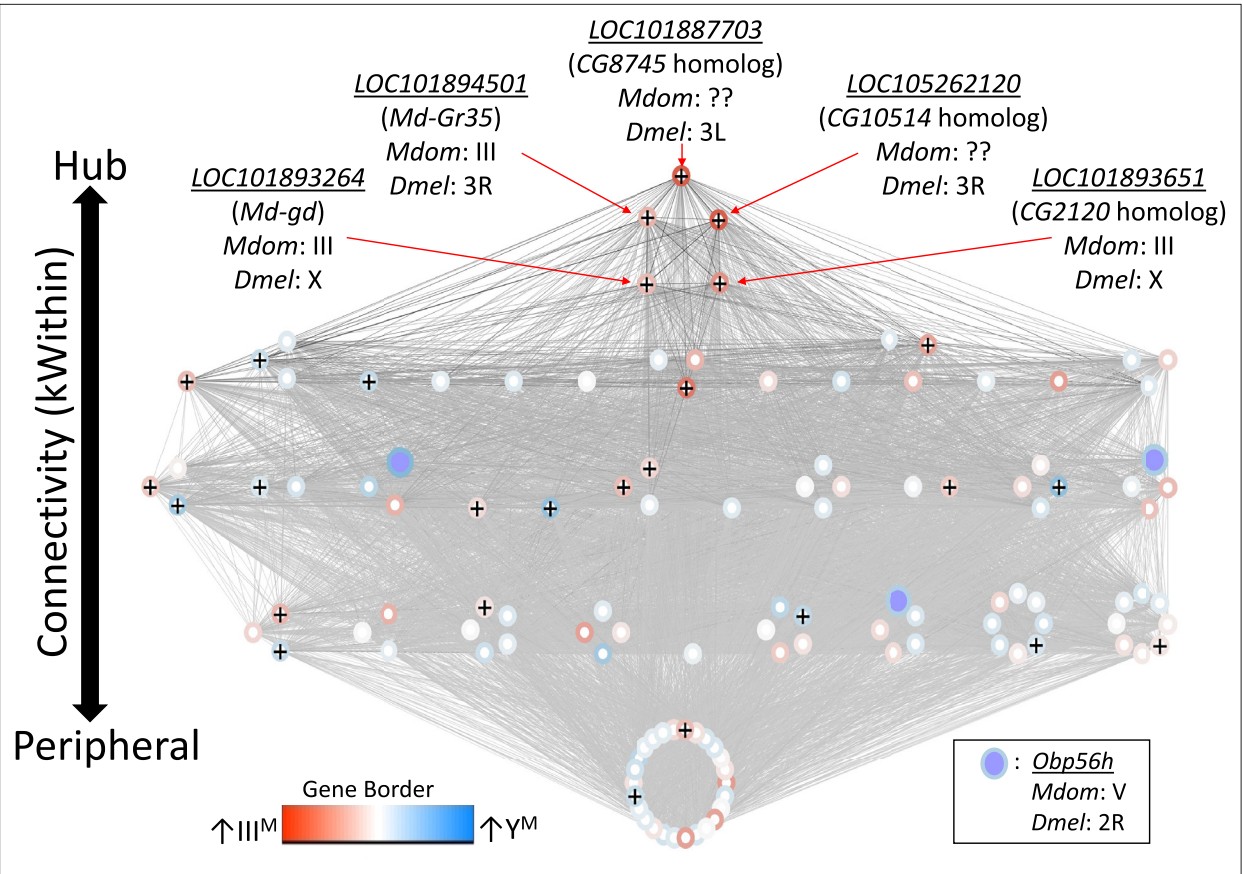

**Figure 3.** Network visualization of the co-expression module that is differentially regulated between III$^M$ and Y$^M$ males. Each circle within the module is a gene, and *Obp56h* genes are indicated with purple fill. Lines represent edge connections between genes. Genes labeled with '+' are within the top 100 most strongly connected to *Obp56h* genes. Genes are ordered from top to bottom according to intramodular connectivity (kWithin), with genes of higher connectivity (i.e. hub genes) on top, and peripheral genes on the bottom. Borders around genes denote log$_2$ fold-change in expression between Y$^M$ and IIII$^M$ male heads, with darker blue borders denoting upregulation in Y$^M$, and darker red borders denoting upregulation in III$^M$. Chromosomal locations in house fly (*Mdom*) and *D. melanogaster* (*Dmel*) are shown for the five hub genes and *Obp56h*.

The online version of this article includes the following figure supplement(s) for figure 3:

**Figure supplement 1.** Measures of intramodular connectivity (kWithin) for individual genes within the gene co-expression module, ranked in descending order.

*Obp56h* gene expression (corresponding to the top 0.55% covarying genes; **Supplementary file 1F**); of the 100 genes with the highest *Obp56h* connection scores, 26 are on chromosome III (Fisher's exact test, p=2.0 × 10$^{-4}$, **Figure 4A**). This enrichment is robust to varying the threshold used to classify a gene as in the top covarying; considering genes with the top 1%, 5%, or 10% covarying expression also resulted in significant enrichment of chromosome III genes (Fisher's exact test, all p<0.05). These results support the hypothesis that *trans* regulatory variants that differ between III$^M$ and the standard chromosome III are responsible for the differential expression of *Obp56h* genes between III$^M$ and Y$^M$ house fly males. We cannot perform the same analysis for the effect of Y$^M$ because only 51 genes have been assigned to the house fly X/Y$^M$ chromosome (**Meisel and Scott, 2018**), limiting our power to detect an excess of genes.

Our network analysis does not ascribe directions to the edges connecting house fly genes, and it is therefore possible that *Obp56h* has *trans* regulatory effects on chromosome III expression. To test this hypothesis, we examined available RNA-seq data from an experiment comparing wild-type *D. melanogaster* with flies in which *Obp56h* had been knocked down (**Shorter et al., 2016**). *Obp56h* is on the right arm of the second chromosome in *D. melanogaster* (2R, or Muller element C), which is homologous to house fly chromosome V (**Foster et al., 1981**; **Weller and Foster, 1993**). House fly chromosome III is homologous to the *D. melanogaster* X chromosome, which is known as Muller

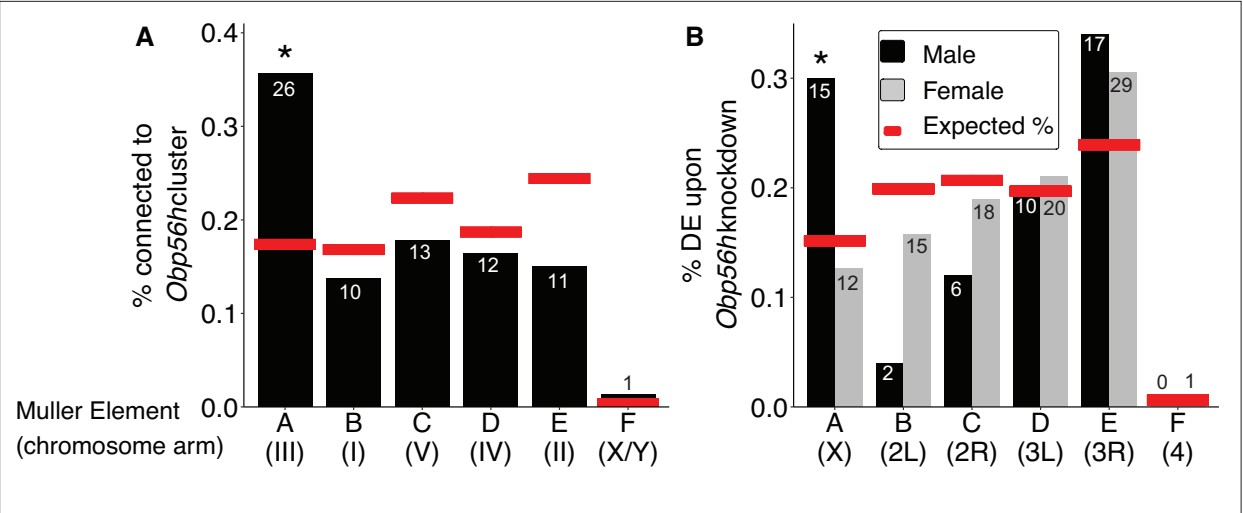

**Figure 4.** *Obp56h* expression is correlated with the expression of genes on Muller Element A. Percent of genes on each chromosome within (**A**) the top 100 genes with the strongest correlated co-expression to the *Obp56h* family in house fly, and (**B**) genes differentially expressed (DE) between *Obp56h* knockdown and control *D. melanogaster* (black bars: males, gray bars: females). Asterisks indicate a significant difference between observed (bars) and expected (red lines) counts of genes on each chromosome compared to all other chromosomes (Fisher's exact test, p<0.05). Numbers within or above bars denote the observed number of genes on a given Muller Element.

The online version of this article includes the following figure supplement(s) for figure 4:

**Figure supplement 1.** Proportions of genes on each chromosome that are differentially expressed in the body (head removed) between *Obp56h* knockdown and control *D. melanogaster* (black bars: males, gray bars: females).

**Figure supplement 2.** Differential expression between male house fly genotypes of genes that are differentially expressed upon *Obp56h* knockdown in *D. melanogaster*.

element A (**Meisel and Scott, 2018**; **Schaeffer, 2018**). The *D. melanogaster* males in the RNA-seq experiment all share the same X chromosome, and only differ in one copy of their second chromosome (which either carries a UAS-RNAi knockdown construct or does not). If *Opb56h* genes have *trans* regulatory effects on element A genes in males, we would expect an excess of DE *D. melanogaster* X chromosome genes in *Obp56h* knockdown flies. Indeed, we found that *Obp56h* knockdown in *D. melanogaster* resulted in an excess of X chromosome DE genes in male head (Fisher's exact test, p=0.011, *Figure 4B*) and body (p=0.038, *Figure 4—figure supplement 1*), but not in either tissue sample in females (Fisher's exact test, both p>0.49). These results suggest that there is male-specific *trans* regulatory control of *D. melanogaster* X-linked genes by *Obp56h*.

We found multiple similarities between house fly and *D. melanogaster* that suggest the genes that regulate and/or are regulated by *Obp56h* are evolutionarily conserved between the two species. Specifically, genes that were downregulated upon knockdown of *Obp56h* in *D. melanogaster* have house fly orthologs that are more downregulated in III[M] male house flies (i.e. lower $\log_2$ fold-change) than expected by chance (p=5.60 × 10[-3], *Figure 4—figure supplement 2A*). However, genes that were upregulated upon *Obp56h* knockdown in *D. melanogaster* were not significantly differentially regulated between Y[M] and III[M] male genotypes, although the observed trend suggests that these genes may be more downregulated in III[M] males than expected (p=0.103, *Figure 4—figure supplement 2B*). The GO term 'response to stress' (GO:0033554) is significantly enriched among genes with strong connection scores with *Obp56h* expression in *M. domestica* and in the list of DE genes in *D. melanogaster* upon *Obp56h* knockdown (*Supplementary file 1G*), providing additional evidence for conserved co-regulation. Altogether, our results suggest that there is an evolutionarily conserved *trans* regulatory feedback loop involving *Obp56h* expression and Muller element A in *Drosophila* and house fly through similar molecular functions.

## Network analysis reveals candidate regulators of Obp56h expression

The house fly co-expression module contains candidate genes and pathways through which *Obp56h* genes, and likely male copulation latency, are regulated. For example, a serine protease gene,

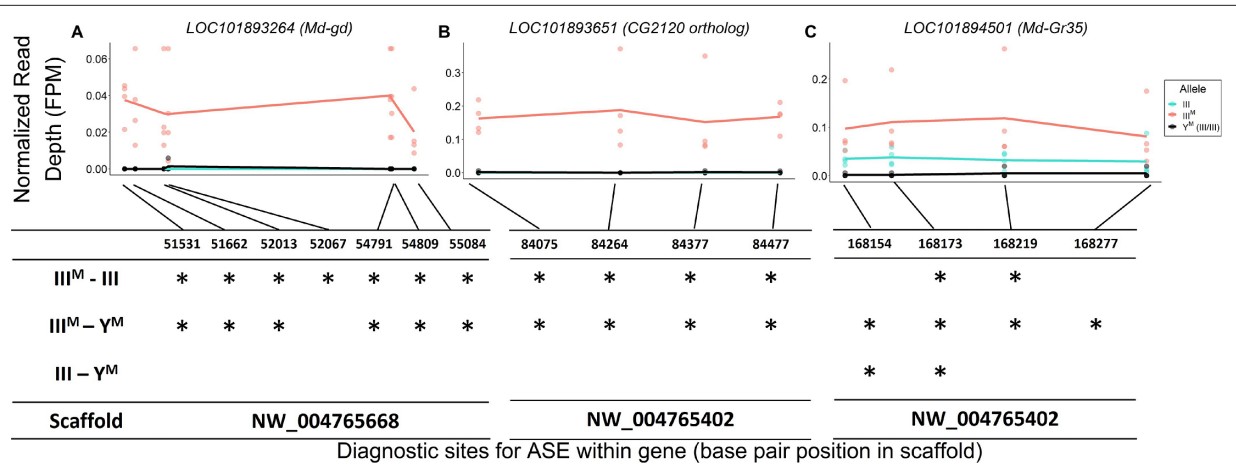

**Figure 5.** Evidence for III^M-biased expression of hub genes on house fly chromosome III. Allele-specific expression (ASE) in (**A**) *LOC101893264* (*Md-gd*), (**B**) *LOC101893651* (*CG2120* ortholog), and (**C**) *LOC101894501* (*Md-Gr35*).The x-axis depicts base pair positions (scaffold coordinates) of the informative single nucleotide polymorphisms (SNPs) that differ between III^M and standard chromosome III alleles. The y-axis and data points depict the read depth of a given allele normalized by the total mapped reads for a given strain-by-experimental batch group combination (FPM = fragments per million). Lines depict mean read depths at each diagnostic site for III (turquoise) and III^M (salmon) alleles in III^M males, and mean read depths at each site for III alleles in Y^M males (black). Tables under each graph mark significant differences (*: p<0.05) in normalized read depths at each diagnostic site for each of three pairwise comparisons: III^M allele vs. III allele in III^M males (III^M-III), III^M allele in III^M males vs. both III alleles in Y^M males (III^M-Y^M), III allele in III^M males vs. both III alleles in Y^M males (III-Y^M).

*LOC101893264*, orthologous to *D. melanogaster gd* (**Konrad et al., 1998**), is among the top 5 hub genes within the co-expression module (**Figure 3**). This gene is predicted to encode a positive regulator of the Toll signaling pathway (**LeMosy et al., 2001**; **Valanne et al., 2011**), suggesting that the *M. domestica* ortholog of *gd* (*Md-gd*) could have an important gene regulatory function within the module via Toll signaling. *Md-gd* is located on chromosome III, and it is upregulated in III^M males (adj. p=0.022). We tested if *Md-gd* is differentially regulated between the III^M chromosome and standard chromosome III by comparing expression in III^M males (i.e. heterozygotes for III^M and a standard chromosome III) with Y^M males that are homozygous for the standard chromosome III. Differential expression of the III^M and III chromosome alleles would implicate *Md-gd* as having a causal effect on *Obp56h* expression. We identified seven polymorphic sites where all RNA-seq reads were mapped to the III^M allele, while no reads were mapped to the standard chromosome III allele in III^M males (**Figure 5A**). At all seven diagnostic single nucleotide polymorphisms (SNPs) in this gene, the III^M allele is significantly more highly expressed than the III allele in III^M males (all p=0.021), and it is more highly expressed than both III alleles in Y^M males at six of seven sites (all p=0.021). Higher expression of the III^M allele is consistent with *cis* regulatory divergence between the III^M and standard chromosome III being responsible for elevated *Md-gd* expression in III^M males. The lack of expression of the III allele in either III^M or Y^M males is consistent with monoallelic gene expression of the III^M allele, although further evidence is required to confirm this hypothesis.

We identified similar evidence of monoallelic gene expression within another hub gene, *LOC101893651*, which is orthologous to *D. melanogaster CG2120* (**Figure 5B**). *LOC101893651* is among the most central genes within the co-expression module (**Figure 3**), and it is strongly upregulated in III^M males (log₂ fold-change: 1.33, p_ADJ = 0.033). *LOC101893651* is found on house fly chromosome III and is predicted to encode a transcription factor. At all four diagnostic sites within *LOC101893651*, the III^M allele is significantly more highly expressed than the III allele in III^M males (all p=0.021), as well as both III alleles in Y^M males (all p≤0.027). Within the WGCNA module, *Obp56h* expression is most strongly correlated with *LOC101893651*, suggesting that *LOC101893651* could encode the transcription factor that is directly responsible for the repression of *Obp56h* expression in III^M males. Consistent with this hypothesis, there is evidence that the protein encoded by *CG2120* binds both upstream and downstream of *Obp56h* in *D. melanogaster* (**Kudron et al., 2018**). Below, we describe experimental results that test the hypothesis that *LOC101893651* (*CG2120*) regulates the expression of *Obp56h* genes.

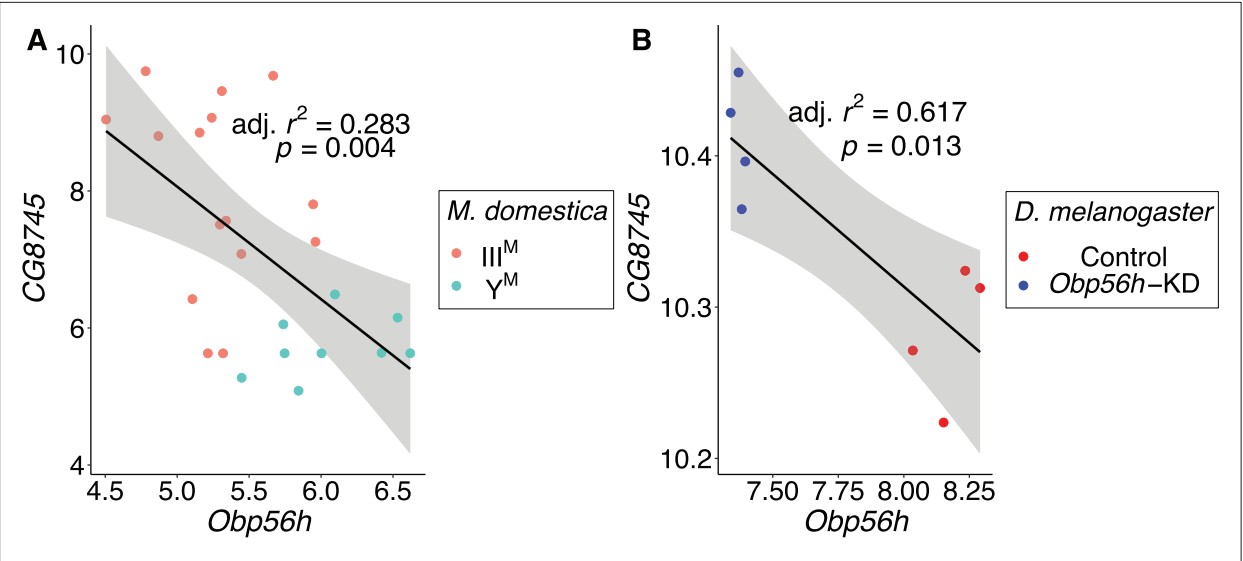

**Figure 6.** *CG8745* and *Obp56h* expression are correlated in male head tissue of both *M. domestica* and *D. melanogaster*. Correlations of gene expression between *Obp56h* (house fly *LOC105261916*) and *CG8745* (*LOC101887703*) in (**A**) house fly male head tissue and (**B**) *D. melanogaster* male head tissue. Values for *D. melanogaster* are from count data as reported in **Shorter et al., 2016**. Linear regression models were used to determine 95% confidence intervals (shaded in gray) summarizing the effect of *Obp56h* expression on *CG8745* expression in each species.

The online version of this article includes the following figure supplement(s) for figure 6:

**Figure supplement 1.** Expression levels in *D. melanogaster* head tissues of *Obp56h* and the hub genes in the co-expression network.

There is also a gustatory receptor gene (*LOC101894501*) that is a hub in the co-expression module and upregulated in III$^M$ males (p$_{ADJ}$ = 0.037). *LOC101894501* (also annotated as *Md-Gr35*) is a homolog of *D. melanogaster Gr98c* (**Scott et al., 2014**). *Md-Gr35* contains four exonic SNPs differentiating the III$^M$ and III chromosomes. Within each III$^M$ strain in each RNA-seq experiment, we observed significantly greater expression of the III$^M$ allele than the standard chromosome III allele at two of the four diagnostic SNP sites (**Figure 5C**). The other two SNPs showed the same pattern of III$^M$-biased expression but were not significant (both p>0.05). The III$^M$ allele in III$^M$ males is also expressed higher than both III alleles in Y$^M$ males, consistent with *cis* regulatory divergence between the III$^M$ and standard chromosome III driving elevated *Md-Gr35* expression in III$^M$ males. However, the standard chromosome III allele is expressed significantly higher in III$^M$ males than Y$^M$ males at two of the four diagnostic SNP sites (**Figure 5C**); we observe the same pattern at the other two sites without significance (p>0.05). Higher expression of the III allele in III$^M$ males than Y$^M$ males suggests that *trans* regulators further increase the expression of *Md-Gr35* in III$^M$ males. This combination of *cis* and *trans* regulatory effects on *Md-Gr35* expression are consistent with the *trans* regulatory loop we hypothesize between *Obp56h* and chromosome III that regulates male copulation latency. Future experiments could determine whether *Gr98c* (*Md-Gr35*) and *Obp56h* do indeed interact and, if so, what pheromonal or other chemical compounds they detect.

In contrast to the aforementioned three hub genes, the most central gene within the module (*LOC101887703*) may be regulated by *Obp56h*. *LOC101887703* is orthologous to *D. melanogaster CG8745*, which is predicted to encode an ethanolamine-phosphate phospho-lyase and is broadly expressed in many *D. melanogaster* tissues (**Chintapalli et al., 2007**). *LOC101887703* is upregulated in III$^M$ males (log$_2$ fold-change: 2.21, p$_{ADJ}$ = 0.016), which could have a causal effect on *Obp56h* DE, be caused by *Obp56h* DE, or neither. In both *D. melanogaster* and house fly, *Obp56h* expression is significantly negatively correlated with the expression of *CG8745* or *LOC101887703*, respectively (**Figure 6**). Directly comparing *CG8745* expression between control and *Obp56h* knockdown *D. melanogaster* yields qualitatively similar results, with *Obp56h* knockdown flies showing greater expression of *CG8745* than controls (Welch's t-test, t=−4.27, p=5.53 × 10$^{-3}$). The negative correlation between *CG8745* and *Obp56h* expression in *D. melanogaster* suggests that *Obp56h* downregulation causes *CG8745* upregulation because *Obp56h* expression was directly manipulated by RNAi in the experiment. The same relationship between *CG8745* and *Obp56h* expression in both house fly and *D.*

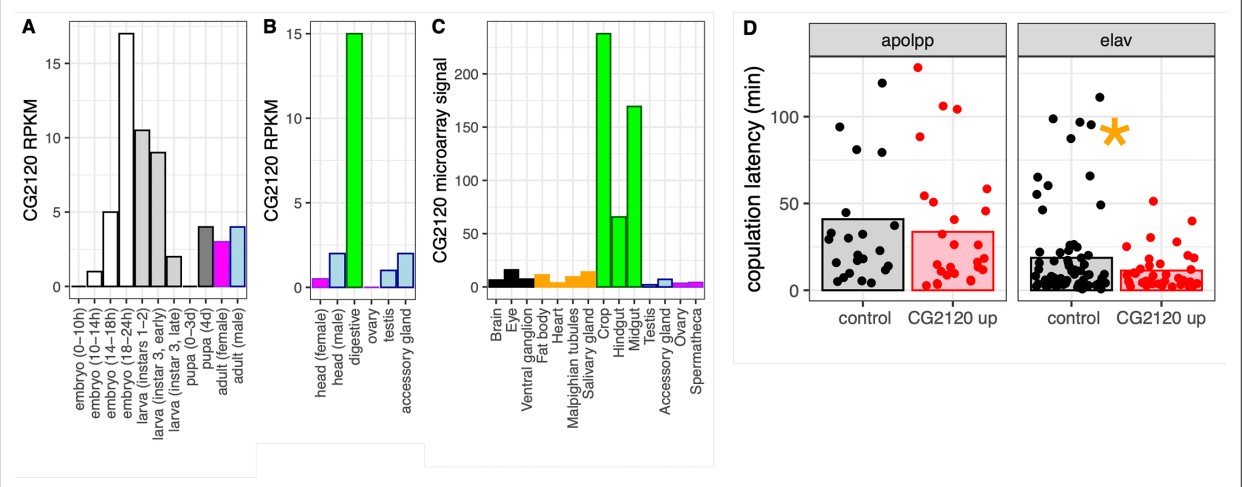

**Figure 7.** CG2120 is broadly expressed and affects copulation latency. Expression levels of *CG2120* are plotted using (**A**) RNA-seq data from developmental stages (*Graveley et al., 2011*), (**B**) RNA-seq data from adult tissues (*Brown et al., 2014*), or (**C**) microarray data from adult tissues (*Chintapalli et al., 2007*). Expression measurements are shown as reads per kilobase per million mapped reads (RPKM) values for RNA-seq data and signal intensity for microarray data (*Öztürk-Çolak et al., 2024*). Bars are colored based on tissue sample categories: embryo (white), larva (light gray), pupa (dark gray), adult female (magenta), adult male (light blue), neuronal (black), digestive tract (green), or other adult tissues (orange). (**D**) Copulation latency in control male *D. melanogaster* and males in which *CG2120* was upregulated (CG2120 up) in fat body cells (using an apolpp Gal4 driver) or neurons (elav driver). Each dot is a single male for which copulation latency was measured. Vertical bars show the estimated effect of male genotype (control or upregulated) from a linear model in which male genotype is a fixed effect and female strain and batch are random effects. The asterisk indicates a significant difference (p<0.05) between control and CG2120 upregulated for a specific tissue driver.

*melanogaster* provides evidence that similar molecular mechanisms may underlie the hypothesized *trans* regulatory effect of *Obp56h* in both species.

We also evaluated the expression of *Obp56h* and the hub genes in the co-expression network module to determine if they are expressed in tissues that could be consistent with the behavioral effects we hypothesize. Specifically, we focused on tissues involved in chemosensation. We confirmed that *Obp56h* is highly expressed in adult *D. melanogaster* head, and we found that expression is most pronounced in the labellum (i.e. proboscis). Notably, the labellum is the only head component in which all five hub genes are expressed (*Figure 6—figure supplement 1*). These findings suggest that the module may function in regulating responses to external chemical stimuli, although similar tissue-specific expression data from *M. domestica* would be needed to provide stronger evidence of this.

## CG2120 expression affects courtship

We investigated the expression of *CG2120*, one of the hub genes in the house fly co-expression network, across *D. melanogaster* development and tissues. Expression of *CG2120* oscillates over the life history of *D. melanogaster*: increasing throughout embryonic development, decreasing with each subsequent larval instar, and then increasing in pupa and in some adult tissues (*Figure 7A–C*). *CG2120* is predominantly expressed in digestive tissues in adults, with much lower expression in nervous tissues (*Figure 7B and C*). Notably, *CG2120* is more highly expressed in male than in female heads (*Figure 7B*), consistent with a regulatory effect on male courtship behavior.

We tested the hypothesis that *CG2120* affects copulation latency in *D. melanogaster* via negative regulation of *Obp56h* expression. To that end, we used CRISPRa to upregulate expression of *CG2120* in fat body cells or neurons of *D. melanogaster* males, using an *apolpp* and *elav* promoter, respectively. We tested if upregulation of *CG2120* reduces copulation latency, as would be expected if *CG2120* negatively regulates *Obp56h*. We used females from two different strains in our assays (CantonS and OregonR). Neither female strain nor the interaction between female strain and male genotype had a significant effect on copulation latency, regardless of the tissues in which expression was induced (*Supplementary file 1H*). There was also not a significant difference in copulation latency between control and *CG2120* upregulated flies when we activated expression in fat body cells (*Supplementary file 1H*, *Figure 7D*). In contrast, there was a significant difference between control

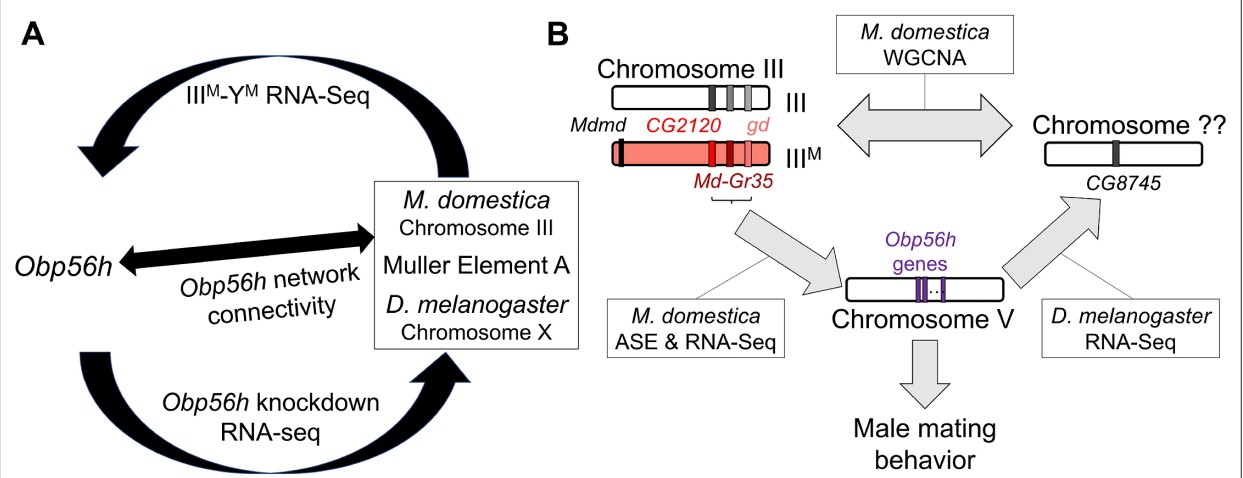

**Figure 8.** Hypothesized relationships between *Obp56h* expression, proto-Y chromosome genotype, and male mating behavior based on house fly and *D. melanogaster* gene expression data. (**A**) Summary of evidence for an evolutionarily conserved *trans* regulatory loop between *Obp56h* and Muller element A (house fly chromosome III and *D. melanogaster* X chromosome). Our hypothesis is based on differential expression between III^M vs. Y^M male house flies, *Obp56h* knockdown vs. control *D. melanogaster*, and network connectivity of *Obp56h* family gene expression within house fly. (**B**) Summary of candidate genes implicated in conserved *trans* regulatory loop. Three of the top five hub genes of module A are located on house fly chromosome III, are negatively correlated with *Obp56h* expression, and exhibit either allele-specific expression (ASE) or show signs of monoallelic gene expression biased toward the III^M allele. Similar correlations between expression measures of *Obp56h* and *CG8745* (*LOC101887703*) in *D. melanogaster* and house fly male head tissue suggest that *Obp56h* regulates *CG8745*, which is the primary hub gene in the weighted gene co-expression network analysis (WGCNA) module that is differentially expressed between III^M and Y^M male house flies. Shared correlations between *Obp56h* expression and copulation latency in both house fly and *D. melanogaster* also suggest that *Obp56h* regulates male fly mating behavior.

and *CG2120* upregulated males when expression was activated in neurons using the *elav* driver, with *CG2120* upregulated males mating faster than controls (***Supplementary file 1H***, ***Figure 7D***). This is consistent with *CG2120* activation in neurons negatively regulating *Obp56h*, which then reduces copulation latency.

## Discussion

We combined analysis of functional genomic data, behavioral experiments, and genetic manipulation to determine the regulatory architecture underlying variation in a courtship trait in flies (***Figure 8***). Our results provide evidence that *Obp56h* expression affects copulation latency in house fly, consistent with its effects in *D. melanogaster* (***Shorter et al., 2016***). Based on their locations on house fly chromosome III, positions as hub genes in a house fly co-expression module (***Figure 3***), and divergent expression between the III^M and standard III chromosomes (***Figure 5***), we hypothesize that *LOC101893651* (*CG2120*), *Md-gd*, and/or *Md-Gr35* (*Gr98c*) negatively regulate *Obp56h* expression (***Figure 8B***). In addition, we hypothesize that *Obp56h* negatively regulates *CG8745* (*LOC101887703*) because knockdown of *Obp56h* causes an increase in *CG8745* expression in *D. melanogaster* (***Figure 6B***). Together, these results suggest that there is a *trans* regulatory loop in which *Obp56h* is both regulated by and regulates expression of genes on fly Muller element A (house fly chromosome III and *D. melanogaster* chromosome X).

We have experimentally validated that *CG2120* affects copulation latency in a way that is consistent with negative regulation of *Obp56h* in *D. melanogaster* neurons (***Figure 7D***). We propose the name *speed date* (*spdt*) for *CG2120* because its expression reduces copulation latency, and we refer to the house fly ortholog (*LOC101893651*) as *Md-spdt*. Future manipulative experiments will help in further evaluating the direction of regulation of the other co-expressed genes.

### Sexual antagonism, pleiotropic constraints, and sexual selection

Repression of *Obp56h* expression reduces copulation latency, which we show is associated with an advantage in male-male competition for female mates (***Figure 2***). The apparently simple correlation between *Obp56h* expression and copulation latency suggests that sexual selection should favor

downregulation of *Obp56h* expression. This raises an important question: why do some males not downregulate *Obp56h* expression to gain a sexually selected advantage? III[M] males do indeed express *Obp56h* at lower levels than Y[M] males (*Figure 1*), demonstrating that downregulation is possible. In addition, experimental repression of *Obp56h* in *D. melanogaster* males also reduces copulation latency relative to wild-type males (*Shorter et al., 2016*), further demonstrating the possibility for downregulation.

We hypothesize that the regulatory architecture associated with *Obp56h* expression creates evolutionary constraints that inhibit selection to reduce expression of *Obp56h* in Y[M] males and *D. melanogaster*. Our analysis of co-expressed genes suggests that *Obp56h* exists within a complex regulatory network, including *trans* feedback, that is conserved between *Drosophila* and house fly (*Figure 8*). This network architecture likely constrains the evolution of *trans* regulators of *Obp56h* and also creates downstream pleiotropic effects of changes to *Obp56h* expression. These pleiotropic effects likely create correlations between traits and weaken the response to selection (*Lande and Arnold, 1983*).

Selective constraints on *Obp56h* expression may arise because the *trans* regulators are evolutionary constrained. We identified three candidate negative regulators of *Obp56h* (*spdt*, *gd*, and *Gr98c*), one or more of which may be constrained in their ability to increase in expression. Importantly, *trans* regulators are predicted to have pleiotropic effects, which could impede the response to selection on traits they affect (*Carroll, 2005*). For example, *gd* is a serine protease involved in Toll signaling (*Konrad et al., 1998*; *LeMosy et al., 2001*). The Toll signaling pathway regulates cellular processes, including embryonic development and immune response to infection (*Belvin and Anderson, 1996*; *Valanne et al., 2011*). Therefore, upregulation of *gd* could have pleiotropic effects that constrain the evolution of *gd* expression.

The expression profile of *spdt* (*CG2120*) suggests that its upregulation in head or neurons might be similarly constrained. Notably, *spdt* is expressed highly in embryos and in the adult digestive tract, but much lower at other developmental stages or tissues, including adult head (*Figure 7A–C*). The high expression in digestive tissues suggests that it is mechanistically possible for *spdt* to evolve higher expression in neurons, which would decrease copulation latency (*Figure 7D*). We hypothesize that constraints on the regulators of *spdt*, or the genes it regulates, likely prevent its upregulation in neurons—specifically neurons in the head that affect copulation latency. Consistent with this prediction, there are 6142 binding locations identified for Spdt from ChIP-seq data (*Kudron et al., 2018*), suggesting that upregulation of *spdt* would have downstream effects on the expression of a large portion of genes in the genome.

We additionally hypothesize that downregulation of *Obp56h* may have negative fitness effects that further prevent the evolution of reduced expression, even though it could be advantageous to males. One deleterious effect may arise from the interaction between *Obp56h* and *Gr98c* (*Md-Gr35*). *Md-Gr35* is a hub gene in the house fly co-expression module containing *Obp56h*, and the expression levels of *Md-Gr35* and *Obp56h* are negatively correlated (*Figure 3*). In addition, we find evidence for both *cis* and *trans* effects on the upregulation of *Md-Gr35* in III[M] males (*Figure 5C*), suggesting a possible feedback on *Md-Gr35* from *Obp56h* downregulation. Obps most typically interact with chemosensory receptors (odorant, ionotropic, and gustatory receptors) in the detection of chemical cues or signals, although they can have other functions as well (*Zhou, 2010*; *Benoit et al., 2017*; *Sun et al., 2018b*). *Md-Gr35* is the only chemosensory receptor assigned to the co-expression module, suggesting that *Md-Gr35* and *Obp56h* are each other's interacting partners. Chemosensory binding proteins and receptors are also known to co-regulate one another, and a negative correlation between the expression of a chemosensory receptor and its interacting binding protein has previously been reported in a pair of genes that modulate male *Drosophila* mating behavior (*Park et al., 2006*). If *Obp56h* serves a sensory detection role in fly heads, then *Gr98c* (*Md-Gr35*) is a promising candidate gene with which it interacts. The interactions between these two genes may constrain selection on *Obp56h* expression levels.

One mechanism by which interactions between *Obp56h*, *Gr98c* (*Md-Gr35*), and other co-expressed genes may create constraints is through the production of cuticular hydrocarbons (CHCs). CHCs are lipid compounds used for both chemical communication and resistance to various environmental stressors, including desiccation (*Chung and Carroll, 2015*). The expression of *Obp56h* is associated with CHC profiles in *D. melanogaster* (*Shorter et al., 2016*). In addition, *LOC105262120*, another hub gene in the network (*Figure 3*), is homologous to *CG10514*, whose expression is correlated with

CHC production in *Drosophila serrata* (*McGraw et al., 2011*). CHCs are often under strong sexual selection across insect systems, with individual or combinations of CHCs serving as important mating cues (*Thomas and Simmons, 2009*; *Berson et al., 2019a*; *Berson et al., 2019b*). The correlation of multiple hub genes with CHC profiles in *Drosophila* provides additional evidence that *Obp56h* expression, and the house fly co-expression module more generally, is related to male mating behavior, and possibly under sexual selection.

Pleiotropic or intersexual conflicts may arise over *Obp56h* expression because CHCs are important for protection against abiotic stressors (*Arya et al., 2010*; *Blomquist and Bagnères, 2010*; *Otte et al., 2018*; *Sun et al., 2018a*), including desiccation resistance (*Lockey, 1988*). Disrupted expression of individual genes responsible for CHC production in *D. melanogaster* can result in significant alterations to both mating behaviors and ecologically relevant phenotypes (*Marcillac et al., 2005*; *Shorter et al., 2016*), which could impede the response to sexual selection via trade-offs with natural selection (*Rowe and Rundle, 2021*). This dual role of CHCs in mating and desiccation resistance suggests that sexual selection on *Obp56h* expression could be pleiotropically constrained by trade-offs with stress response. Our GO enrichment analysis on both the house fly and *D. melanogaster* RNA-seq data also revealed that *Obp56h* expression is correlated with the expression of genes involved in general stress responses, supporting this hypothesis. Pleiotropic constraints on *Obp56h* expression (because of correlated changes in CHCs) could therefore reduce the response to selection on male copulation latency.

Additional pleiotropic constraints are possible because Obps have myriad functions beyond chemical detection (*Findlay et al., 2008*; *Arya et al., 2010*; *Benoit et al., 2017*; *Sun et al., 2018a*). *Obp56h* expression specifically has been shown to affect mating behavior, avoidance of bitter tastants, and the expression of genes related to immune response and heat stress (*Swarup et al., 2014*; *Shorter et al., 2016*). Moreover, single base pair changes at the *Obp56h* locus in *D. melanogaster* cause sex-specific effects on a wide range of fitness-related traits, including heat resistance (*Mokashi et al., 2021*). Future studies should aim at determining whether *Obp56h* expression in III^M and Y^M male house flies is also associated with other phenotypes, such as CHC profiles, desiccation resistance, or tolerance to other environmental stressors.

In general, our results provide evidence that pleiotropy can reduce the response to sexual selection by creating genetic covariation among unrelated traits (*Fitzpatrick, 2004*; *Chenoweth and McGuigan, 2010*). Specifically, selection on *Obp56h* expression and male courtship behavior could be weakened by trade-offs between courtship behavior and stress response. In addition, the regulatory architecture underlying *Obp56h* expression likely creates additional pleiotropic constraints that could impede selection on copulation latency. One expected consequence of these pleiotropic constraints is that there will be genetic variance for sexually selected traits in natural populations (*Kirkpatrick and Ryan, 1991*; *Turelli and Barton, 2004*; *Johnston et al., 2013*; *Heinen-Kay et al., 2020*), which we observe in the form of variation in copulation latency between III^M and Y^M males.

## Sexually antagonistic alleles on Y chromosomes

Our results suggest that III^M males may have overcome the pleiotropic constraints associated with *Obp56h* expression via the Y-linkage of at least one *trans* regulator of *Obp56h*. We identified three candidate *trans* regulators of *Obp56h* on the house fly III^M chromosome (*Md-spdt*, *Md-gd*, and *Md-Gr35*), all of which could be inhibitors of Obp56h expression (*Figure 8*). This is consistent with the previous work suggesting that male-beneficial *trans* regulatory alleles could be important for the fitness effects of the house fly III^M chromosome (*Adhikari et al., 2021*). More generally, *trans* regulation of autosomal and X chromosome expression appears to be an important feature of the phenotypic and fitness effects of Y chromosomes (*Chippindale and Rice, 2001*; *Lemos et al., 2008*; *Lemos et al., 2010*; *Brown et al., 2020*; *Morris et al., 2020*; *Kawamoto et al., 2021*; *Sandkam et al., 2021*).

We hypothesize that the pleiotropic constraints associated with the regulation of *Obp56h* expression are a source of sexual antagonism. As described above, deleterious fitness effects could result from both pleiotropic effects of upregulation of the negative regulators of *Obp56h* and downstream effects of the downregulation of *Obp56h*. Regardless of the cause, these fitness costs are likely to affect both males and females. However, downregulation of *Obp56h* confers a fitness benefit to males that may offset any fitness costs. III^M males can realize this fitness benefit because the III^M chromosome carries at least one negative *trans* regulator of *Obp56h*. Therefore, our results provide evidence that

sexual antagonism can arise via conflicts between sexual selection in males and opposing pleiotropic effects in females (*Lande, 1980*; *Fitzpatrick, 2004*; *Mank et al., 2008*). It also appears that the intersexual conflict over *Obp56h* expression was (at least partially) resolved with the Y-linkage of the *trans* regulators in III[M] males.

Upregulation of the III[M] alleles of *Md-spdt*, *Md-gd*, and/or *Md-Gr35* alone cannot fully resolve the intersexual conflict over *Obp56h* regulation for at least two reasons. First, Y[M] males do not realize the fitness benefit because they do not carry a III[M] chromosome. Therefore, Y[M] males remain evolutionarily constrained in their expression of *Obp56h*, suggesting that other fitness benefits are responsible for the maintenance of the Y[M] chromosome. Previous work suggests that temperature-dependent fitness differences could be responsible—specifically Y[M] males have higher fitness at colder temperatures (*Feldmeyer et al., 2008*; *Delclos et al., 2021*). This suggests that the III[M] courtship advantage may be temperature-sensitive, which could create context-dependent fitness effects that favor Y[M] males in specific environments. This is consistent with the prediction that genotype-by-environment interactions can maintain variation in sexually selected traits across heterogeneous environments (*Kokko and Heubel, 2008*). In contrast to that prediction, we failed to detect effects of temperature on the III[M] courtship advantage (*Figure 2*), although we did not comprehensively test for environmental effects. Additional work could evaluate whether trade-offs across environments create opportunities for Y[M] males to have courtship advantages in specific contexts.

A second reason that the intersexual conflict over *Obp56h* regulation is not fully resolved is that female house flies can carry the III[M] chromosome. The male determining gene on the Y[M] and III[M] chromosome (*Mdmd*) is a negative regulator of *Md-tra* splicing, and there is a dominant allele (*Md-tra[D]*) that is desensitized to the effects of *Mdmd* (*Hediger et al., 2010*; *Sharma et al., 2017*). This allows female house flies to carry Y[M] or III[M] if they also carry *Md-tra[D]*. Females with III[M] and *Md-tra[D]* in their genotype may suffer a fitness cost from upregulation of *Md-spdt*, *Md-gd*, and *Md-Gr35* and/or downregulation of *Obp56h*. Therefore, even though the putatively sexually antagonistic alleles are Y-linked, they can still be carried by females, leaving the conflict unresolved. Consistent with this interpretation, previous work has determined that sexually antagonistic fitness effects of the house fly proto-Y chromosomes could contribute to the maintenance of the Y[M]-III[M] polymorphism within natural populations (*Meisel et al., 2016*; *Meisel, 2021*). Our results provide a mechanism by which the III[M] chromosome could confer sexually antagonistic effects.

## Commonalities across sex chromosome evolution

Our results have two additional implications for our understanding of the evolution of sex chromosomes. First, our results provide evidence that gene duplication is important for the acquisition of genes with male-specific functions during the evolution of Y chromosomes (*Koerich et al., 2008*). The most central gene in the co-expression module, *LOC101887703*, has a paralog in the house fly genome (*LOC101890114*) that is predicted to be on chromosome III. These two genes are homologous to *D. melanogaster CG8745*, which we hypothesize is negatively regulated by *Obp56h* based on conserved correlated expression in *D. melanogaster* and house fly (*Figure 6*). The two transcripts encoded by *LOC101887703* and *LOC101890114* are <1% diverged in their nucleotide sequences, suggesting a recent duplication event. *CG8745* is broadly expressed in many *D. melanogaster* tissues (*Chintapalli et al., 2007*), and broadly expressed genes often give rise to paralogs with sex-specific expression (*Meisel et al., 2009*).

We hypothesize that one of the two paralogs (*LOC101887703* or *LOC101890114*) has evolved a male-specific function, which resolved an intersexual conflict associated with the coregulation of *CG8745* and *Obp56h*. It is not clear which gene is ancestral and which is derived. However, a derived paralog with sex-specific expression is consistent with duplication of a broadly expressed gene to resolve an intersexual conflict (*Connallon and Clark, 2011*; *Gallach and Betrán, 2011*; *VanKuren and Long, 2018*). Those types of duplications often involve genes on sex chromosomes—either when a broadly expressed X-linked gene gives rise to an autosomal gene with sex-specific expression, or when a gene is duplicated onto the Y chromosome and evolves male-specific expression (*Betrán et al., 2002*; *Emerson et al., 2004*; *Koerich et al., 2008*; *Meisel et al., 2009*; *Hall et al., 2013*; *Mahajan and Bachtrog, 2017*; *Ricchio et al., 2021*). Consistent with this evolutionary trajectory, the only identified differences between the house fly Y[M] and X chromosomes thus far have been

autosome-to-Y duplications (*Meisel et al., 2017*). Future work could address a potential sexually dimorphic subfunctionalization of the two *CG8745* paralogs in the house fly genome.

Second, our results suggest that intersexual conflict may have been an important factor in the convergent evolution of Muller element A into a sex chromosome in both *Drosophila* and house fly. Muller element F was the X chromosome of the most recent common ancestor of *Drosophila* and house fly, and element A became the *Drosophila* X chromosome after the divergence with most other flies (*Vicoso and Bachtrog, 2013*). Element A also recently became a sex chromosome in house fly when it acquired an *Mdmd* gene, creating the III^M chromosome (*Meisel et al., 2017*; *Sharma et al., 2017*; *Son and Meisel, 2021*). Element A has become a sex chromosome in at least two other dipteran lineages: *Glossina* (tsetse flies) and *Anopheles* mosquitoes (*Pease and Hahn, 2012*; *Vicoso and Bachtrog, 2015*), raising the possibility that element A is primed to be recruited as a sex chromosome. Similar convergent sex-linkage of the same chromosomal region has been observed in vertebrates (*O'Meally et al., 2012*; *Furman and Evans, 2016*; *Ezaz et al., 2017*), which could be explained by the same gene independently acquiring a sex determining allele in multiple independent lineages (*Takehana et al., 2014*).

We hypothesize that element A may be convergently recruited to be a sex chromosome because the *trans* regulatory connections with *Obp56h* create sexually antagonistic effects related to male mating behavior. This differs from the hypothesized cause of convergent evolution of sex chromosomes in vertebrates, which is based on the expectation that some chromosomes contain an excess of genes that are predisposed to become sex determiners. Sexually antagonistic alleles are expected to be an important selective force in the formation of new sex chromosomes because sex-limited inheritance can resolve the intersexual conflict (*van Doorn and Kirkpatrick, 2007*; *Roberts et al., 2009*). Our results suggest that an enrichment of genes involved in a regulatory network with sexually antagonistic effects could promote the sex-linkage of the same chromosome in distantly related species without convergent evolution of a master sex determiner.

## Conclusions

We have identified *Obp56h* expression as a candidate mechanism implicated in regulating courtship performance between Y^M and III^M house flies. We further identified multiple candidate *trans* regulators of *Obp56h* on the III^M chromosome, one of which (*spdt*) we experimentally verified in *D. melanogaster*. Our results demonstrate how the *trans* regulators of gene expression could have sexually antagonistic effects, which are resolved via the Y-linkage of those *trans* factors. This is the first time, to our knowledge, that a mechanism has been ascribed to the observation that the fitness effects of Y chromosomes can manifest via *trans* effects on autosomal genes, which affect male courtship and other sexually selected traits (*Chippindale and Rice, 2001*; *Morris et al., 2020*; *Kawamoto et al., 2021*; *Sandkam et al., 2021*).

## Materials and methods
### RNA-seq differential gene expression analysis

We analyzed published RNA-seq data from *M. domestica* male heads (NCBI Gene Expression Omnibus accessions GSE67065, GSE126685, and GSE126689, shown in *Supplementary file 1A*). The RNA-seq data include 9 Y^M and 15 III^M samples (*Meisel et al., 2015*; *Son et al., 2019*). We assigned RNA-seq reads to house fly transcripts from genome assembly v2.0.2 and annotation release 102 (*Scott et al., 2014*) using kallisto in single-end read mode (*Bray et al., 2016*). All RNA-seq reads were single-end, and we set the average fragment length to 300 bp and standard deviation to 30 bp for all samples. Our expression analysis was performed at the level of transcripts (as opposed to genes), which should not affect our conclusions because all of our focal genes each produce one annotated transcript.

We tested for DE transcripts between males with a Y^M chromosome and males with a III^M chromosome using a combination of DESeq2 (*Love et al., 2014*), sva (*Leek et al., 2012*), and limma (*Ritchie et al., 2015*). We only included transcripts that passed an initial threshold filter of 0.5 counts per million in at least four samples. Read counts for each of those transcripts were normalized by variance stabilizing transformation in DESeq2. To remove batch effects across data sets, we used the sva package to identify and estimate surrogate variables that adjust for latent sources of variation (e.g. batch effects). To identify DE transcripts between Y^M and III^M males, we used the lmFit() function in

limma to fit a linear model comprised of male type ($Y^M$ vs. $III^M$) and our surrogate variables as fixed effects, and read counts as the response variable. We then computed contrasts between male types and calculated test statistics using the eBayes() function. Transcripts below an FDR-adjusted p-value ($p_{ADJ}$) of 0.05 were categorized as DE (*Benjamini and Hochberg, 1995*).

## Weighted gene co-expression network analysis

We used WGCNA to identify modules of house fly transcripts whose expression correlates with male type ($Y^M$ or $III^M$) on normalized read count data that were adjusted for batch effects in sva (*Langfelder and Horvath, 2008*). For all pairs of transcripts with variable expression across samples, we calculated Pearson's correlation coefficient across all samples. We created an unsigned correlation matrix and adjusted the soft-threshold value (β) to which among-transcript covariances are exponentially raised to ensure a scale-free topology (this resulted in β=7), thereby creating a weighted network of gene expression. An unsigned matrix allows us to identify connected transcripts whose expression is either positively or negatively correlated. Within this topological overlapping network (*Li and Horvath, 2007*), transcripts were hierarchically clustered, and modules were identified based on the degree of similarity among transcripts. We used a merging threshold of 0.2, with a minimum module size of 30 and a mean connectivity threshold of greater than or equal to 0.7. We used the default parameters of WGCNA for the rest of the analyses. We then correlated module eigengene values for a given module across samples via Pearson's correlation and identified modules differentially regulated between male types at FDR-adjusted p<0.05.

To visualize WGCNA genetic covariance results among modules significantly associated with male type, we exported final co-expression networks to Cytoscape (*Shannon et al., 2003*). We attached information on $log_2$ fold-change in expression between $III^M$ and $Y^M$ males, as well as chromosomal location, to the network as metadata so this information could be visualized. To identify transcripts that may have more central functions within and across our significant modules, we ranked transcripts in descending order based on intramodular connectivity (kWithin), calculated in WGCNA. Hub genes identified by intramodular connectivity are generally functionally important transcripts within a module (*Langfelder et al., 2013*).

We further analyzed among-transcript connections involving a family of odorant binding protein genes (*Obp56h*). Specifically, to identify transcripts that may regulate or be regulated by genes within the family, we calculated a 'connection score' $C_i$ for every transcript i as follows:

$$C_i = \sum a_{ij} \times |F_j|,$$

where $a_{ij}$ represents the adjacency (the Pearson's correlation coefficient raised to the soft-threshold power β) between transcript i and *Obp56h* gene j, and $F_j$ represents the $log_2$ fold-change in expression between $III^M$ and $Y^M$ males for *Obp56h* gene j. This weighted product ensured that connections with *Obp56h* genes that are more DE between male types were prioritized in calculating a transcript's connection score. Transcripts were then ranked by $C_i$ to identify candidate genes that may be strongly tied to *Obp56h* expression. Transcripts with the 100 highest *Obp56h* connection scores were classified as 'central genes'. We tested for chromosomal enrichment among these central genes using Fisher's exact tests (comparing the number of central and non-central genes on a focal chromosome with the number of central and non-central genes on all other chromosomes) to determine whether the expression of *Obp56h* genes (which are all located on the *M. domestica* chromosome V) might be involved in *trans* regulation with genes located on the $III^M$ proto-Y chromosome.

## GO enrichment analysis

To identify GO classes and molecular pathways that are enriched among DE transcripts, across co-expression modules identified in WGCNA, or among central genes co-expressed with *Obp56h* genes, we used the BiNGO plug-in within Cytoscape (*Maere et al., 2005*). We identified *D. melanogaster* orthologs for each house fly gene within a given gene list via NCBI blastx best hits (with default parameters) and used the *D. melanogaster* gene name as input (*Adhikari et al., 2021*). We identified GO terms that are significantly enriched in BiNGO for biological processes, cellular components, and molecular function.

## Allele-specific expression analysis

We tested for differential expression of house fly chromosome III genes between the allele on the III$^M$ chromosome and the allele on the standard third (III) chromosome in III$^M$ males. To do so, we followed methods as in previous studies (*Meisel et al., 2017*; *Son and Meisel, 2021*), which used the Genome Analysis Toolkit (GATK) best practices workflow for SNP calling to identify sequence variants in our RNA-seq data (*McKenna et al., 2010*). We focused our analysis on libraries that were sequenced from head tissue of male house flies that comprise a CS genetic background (*Meisel et al., 2015*; *Son et al., 2019*; *Adhikari et al., 2021*). We used STAR (*Dobin et al., 2013*) to align reads from a total of 30 head libraries (15 III$^M$ and 15 Y$^M$ libraries) to the house fly reference genome (Musca_domestica-2.0.2). We then followed the same methods and applied the same parameters as we have done previously to identify SNPs and genotype individual strains (*Meisel et al., 2017*; *Son and Meisel, 2021*). We performed separate joint genotyping for each house fly strain within a given experiment (a total of 4 III$^M$ and 4 Y$^M$ strain-by-experimental batch combinations).

We use the following approach to differentiate between III$^M$ and standard chromosome III alleles. We first identified SNPs in the exonic regions of the top 'hub' genes within a WGCNA module that mapped to house fly chromosome III. We selected SNPs in those genes that are heterozygous in III$^M$ males and homozygous in Y$^M$ males. We used the genotype of these SNPs in Y$^M$ males (which possess two standard third chromosome alleles) to determine the standard chromosome III allele. The allele not found in Y$^M$ genotypes was assigned to the III$^M$ chromosome. We also identified positions where III$^M$ males appear monoallelic for an allele not found in Y$^M$ males. These positions that exhibit a complete bias for a III$^M$ allele are suggestive of monoallelic expression of the III$^M$ allele (i.e. no expression from the III allele).

We tested for differences in expression of III$^M$ and standard chromosome III alleles by following best practices for comparing allele-specific expression (*Castel et al., 2015*). First, for each strain-by-experimental batch combination, we calculated the normalized read depth at each variable site as the number of mapped reads at that site divided by the total number of mapped reads throughout the genome. At each variable site, we used Wilcoxon rank sum tests to make three different pairwise comparisons per site. First, we compared normalized read depths between III$^M$ and III alleles in III$^M$ males (III$^M$-III). Second, we compared the read depths of the III$^M$ allele in III$^M$ males with the normalized read depth of both III alleles in Y$^M$ males (III$^M$-Y$^M$). Third, we compared the read depths of the III allele in III$^M$ males with the normalized read depth of both III alleles in Y$^M$ males (III-Y$^M$). We set a threshold of significance at $p<0.05$ for all comparisons.

## *D. melanogaster* RNA-seq and microarray data analysis

We analyzed RNA-seq results reported in a previous study (*Shorter et al., 2016*) to determine how knockdown of *Obp56h* affects gene expression in *D. melanogaster*. *Shorter et al., 2016*, identified DE genes between *Obp56h* knockdown and control samples. This analysis was done separately in males and females, and in separate tissue samples within a given sex (head or the remaining body). We conducted GO enrichment analysis, as described above, on the list of DE genes in *D. melanogaster* male head tissue upon *Obp56h* knockdown.

We tested if an excess of DE genes (between *Obp56h* knockdown and controls) are found on the *D. melanogaster* X chromosome, which is homologous to house fly chromosome III (*Foster et al., 1981*; *Weller and Foster, 1993*). This chromosome is known as Muller element A across flies (*Meisel and Scott, 2018*; *Schaeffer, 2018*). *Obp56h* is located on *D. melanogaster* chromosome 2R (Muller element C), which is homologous to house fly chromosome V. We used Fisher's exact tests (comparing the number of X and non-X chromosome genes that are DE in a given tissue within a given sex with the number of X and non-X chromosome genes that are not DE) to determine whether *Obp56h* knockdown in *D. melanogaster* results in the disproportionate differential expression of X chromosome genes in male heads, male bodies, female heads, or female bodies.

We also tested if the same genes are DE between III$^M$ vs. Y$^M$ house flies and *Obp56h* knockdown vs. control *D. melanogaster*. Using NCBI blastx best hits, we identified 20 *M. domestica* transcripts that are orthologous to *D. melanogaster* genes that are DE upon knockdown of *Obp56h* (11 matches to upregulated *D. melanogaster* genes, and 9 matches to downregulated *D. melanogaster* genes). This list represents 40% of the 50 *D. melanogaster* DE genes. We compared the mean log$_2$ fold-changes between Y$^M$ and III$^M$ house fly males for those 20 genes to 10,000 random subsets of log$_2$ fold-change

values taken from our data (10,000 subsamples without replacement of 11 genes to test for an excess of positive $\log_2$ fold-change values, and 10,000 subsamples of nine genes to test for an excess of negative $\log_2$ fold-change values; see Additional Files for R script). We assessed significance by calculating the proportion of replicated subsamples that generated a mean $\log_2$ fold-change value more extreme than our observed mean.

We additionally obtained gene expression measurements from microarray and RNA-seq data collected from various larval, pupal, and adult *D. melanogaster* tissue samples (*Supplementary file 1B*). Microarray data were sampled from FlyAtlas expression measurements of larval central nervous system, adult head, adult eye, and adult brain (*Chintapalli et al., 2007*). RNA-seq data were sampled from larval, pupal, or adult antennae (*Shiao et al., 2013*; *Menuz et al., 2014*; *Pan et al., 2017*; *Mohapatra and Menuz, 2019*) and adult proboscis (*May et al., 2019*). We only included samples from wild-type flies, and we used the published expression estimates (e.g. microarray signal intensity, transcripts per million, reads per kilobase per million mapped reads [RPKM]) measured at either the gene or transcript level. When data from multiple replicates were available, we calculated the mean expression level across all replicates for each gene or transcript. The range and distributions of expression levels varied across tissue samples because the data were collected using different methodologies. To compare across tissue samples, we calculated a normalized expression level for each gene or transcript in each tissue sample by dividing by the mean value across all genes or transcripts in that tissue sample. Any gene or transcript with an expression value of 0, detected in <4 replicates of the microarray data, or that failed to pass threshold in an RNA-seq data set (i.e. status not OK) was excluded. We further extracted expression measurements from genes identified as hubs within the WGCNA co-expression network (see above). If a hub gene had multiple annotated transcripts, and expression was measured for transcripts, we calculated the gene expression level as the mean across all annotated transcripts.

We obtained measurements of the expression level of *CG2120* from FlyAtlas microarray and modENCODE RNA-seq data sampled at multiple developmental stages and across tissues (*Chintapalli et al., 2007*; *Graveley et al., 2011*; *Brown et al., 2014*). The FlyAtlas microarray data are the average signal intensities across all probes for the gene, and the modENCODE data are reported as RPKM, both of which were obtained from FlyBase (*Öztürk-Çolak et al., 2024*).

## ADD calculations for flies used in mating assays

We used the following formula to estimate accumulated degree day (ADD) of flies reared at different developmental temperatures for competitive and single-choice mating assays:

$$ADD = (T_D - T_t) \times d$$

where $T_D$ is the developmental temperature of a given fly and $T_t$ is the threshold developmental temperature of 12.4°C, which was calculated for house flies in *Wang et al., 2018*. Based on this value, male flies that developed at 18°C and 29°C for competitive mating assays had ADD ranges of 33.6–39.2 and 66.4–83 degree days (dd), respectively. While these ADD ranges do not overlap between developmental temperature treatments, these ranges represent time points during which adult house flies are mating in colony cages.

ADD male house flies developed at 18°C (competitive mating assays)
ADD = (18°C–12.4°C) $\times$ (6d)=33.6 dd
ADD = (18°C–12.4°C) $\times$ (7d)=39.2 dd
ADD male house flies developed at 29°C (competitive mating assays)
ADD = (29°C–12.4°C) $\times$ (4d)=66.4 dd
ADD = (29°C–12.4°C) $\times$ (5d)=83 dd

Male flies that developed at 22°C and 29°C in our single-choice mating assays had ADD ranges of 96–105.6 and 99.6–116.2 dd, respectively. Female flies used in these assays had ADD ranges of 100.8–113.4 dd. These ranges overlap with our estimate of ADD ranges for *D. melanogaster* adults used in *Shorter et al., 2016*: those flies had ADD ranges of 45–105 dd based on the reported age of assayed adults (3–7 days), and an estimated $T_t$ of 10°C (*Loeb and Northrop, 1917*; *Bliss, 1926*; *Koštál et al., 2016*).

ADD male house flies developed at 22°C (single-choice assays)

ADD = (22°C–12.4°C) × (10d)=96 dd
ADD = (22°C–12.4°C) × (11d)=105.6 dd
ADD male house flies developed at 29°C (single-choice assays)
ADD = (29°C–12.4°C) × (6d)=99.6 dd
ADD = (29°C–12.4°C) × (7d)=116.2 dd
ADD female house flies developed at 25°C (single-choice assays)
ADD = (25°C–12.4°C) × (8d)=100.8 dd
ADD = (25°C–12.4°C) × (9d)=113.4 dd
ADD fruit flies assayed in *Shorter et al., 2016*
*ADD* = (25°C–10°C) × (3d)=45 dd
*ADD* = (25°C–10°C) × (7d)=105 dd

## Competitive courtship assays

We performed competitive courtship experiments in which two different house fly males were combined with a single female, and we recorded the 'winning' male (i.e. the one who mated with the female), similar to what was done previously (*Hamm et al., 2009*). In these experiments, we used the same two house fly strains as in *Hamm et al., 2009*: a III$^M$ strain called CS and a Y$^M$ strain called IsoCS. These two strains have a common genetic background (CS), and only differ in which proto-Y chromosome they carry. Both strains are represented in the RNA-seq data we analyzed (*Meisel et al., 2015*; *Son et al., 2019*), and IsoCS was also included in a previous RNA-seq study comparing the effects of proto-Y chromosome and temperature on gene expression (*Adhikari et al., 2021*). Our experiment differed from previous work because we reared larvae from each strain at either 18°C and 29°C, whereas *Hamm et al., 2009*, worked with flies raised at 28°C. We used the same larval wheat bran diet as done previously, and we fed adults an ad libitum supply of water and an ad libitum 1:1 mixture of dry-milk:sugar. This is also the same diet and rearing protocol used for the flies in the RNA-seq data sets that we analyzed (*Meisel et al., 2015*; *Son et al., 2019*). Male flies were isolated from females within ~1 hr of eclosion, and each sex was kept separately to ensure that flies had not mated prior to the experiment.

We carried out two distinct competitive courtship experiments: (1) inter-strain competition between males with different genotypes (i.e. Y$^M$ vs. III$^M$) that were reared at the same temperature (363 successful mating trials out of 490 total attempts across 27 experimental batches); and (2) intra-strain competition between males with the same genotype that were reared at different temperatures (104 successful mating trials out of 129 total attempts across 7 batches). When we competed flies with different genotypes raised at the same temperature, all males were aged 4–6 days post pupal emergence. When we compared flies with the same genotype raised at different temperatures, 29°C males were aged 4–5 days post emergence and 18°C males were aged 6–7 days post emergence. We aged flies from the colder temperature for more days than flies from the warmer temperature because developmental rate is positively correlated with developmental temperature in flies (*Atkinson, 1996*). The ages we selected ensure that all males were physiologically capable of mating, while also sampling flies at similar physiological ages across experiments.

The two males in each experiment were labeled using red and blue luminous powder (BioQuip) by shaking the flies in an 8 oz paper cup. The color assigned to males was switched in each successive batch (i.e. blue Y$^M$ and red III$^M$ in one batch, and then red Y$^M$ and blue III$^M$ in the next batch). In addition, we included the genotype or developmental temperature of the blue-colored male as a fixed effect in our statistical analysis (see below), which provides an additional control for color.

For each replicate of the competitive courtship assay, we placed the two different males in a 32 oz transparent plastic container, along with a single virgin female. Each plastic container also contained a 1:1 mixture of dry milk:sugar in a 1 oz paper cup and water in a glass scintillation vial plugged with a cotton roll. Virgin females from the LPR strain (*Scott et al., 1996*) raised at 25°C were used for all combinations of males. The LPR strain has a different genetic background than the males used in the assay, minimizing any effects of co-adaptation between females and a particular subset of males. All flies were transferred into the mating containers using an aspirator and without anesthesia. All matings were performed in a 25°C incubator because copulation latency is too long for experimentally tractable measurement at lower temperatures. The color (i.e. genotype) of the first male to mate was recorded, as well as the time to mate.

We used the glmer() function in the lme4 package in R (*Bates et al., 2015*) to test for the effects of genotype and temperature on male mating success. First, to test the effect of genotype, we constructed a logistic regression model as follows:

$$W \sim G_B + T + G_B \times T + b,$$

with developmental temperature (T), genotype of the blue male ($G_B$), and their interaction as fixed effects. Experimental batch (b) was modeled as a random effect, with the winning male (W: CS or IsoCS) as a response variable. We then assessed significance of fixed effects (type II sum of squares) using the Anova() function in the car package in R (*Fox et al., 2013*). To test for the effect of temperature on mating success, we similarly constructed a logistic regression model as follows:

$$W \sim G + T_B + G \times T_B + b,$$

with genotype (G), developmental temperature of the blue male ($T_B$), and their interaction as fixed effects, experimental batch (b) as a random effect, and the winning male (W: 18°C or 29°C) as a response variable. We then assessed significance of fixed effects (type II sum of squares) using the Anova() function in the car package in R (*Fox et al., 2013*).

## Single-choice courtship assays

We performed experiments to measure copulation latency, or the amount of time elapsed before mating, according to male type ($Y^M$ or $III^M$). In these experiments, we used the same IsoCS ($Y^M$) and CS ($III^M$) strains as above and in *Hamm et al., 2009*. We also tested one other strain from each genotype. CSrab ($III^M$) was created by backcrossing the $III^M$ chromosome from the rspin strain isolated in New York onto the CS background (*Shono and Scott, 2003*; *Son et al., 2019*). CSaY ($Y^M$) was created by backcrossing the $Y^M$ chromosome from the aabys genome reference strain onto the CS background (*Scott et al., 2014*; *Meisel et al., 2015*). Virgin females used in the assays were all from the LPR strain (*Scott et al., 1996*), which has a different genetic background than all males tested. In addition, we also assayed LPR males to determine how copulation latencies of $III^M$ and $Y^M$ males compare to those of males from the same genetic background as the females.

We first attempted to test flies reared at the same temperatures as in our competitive courtship assays (18°C and 29°C), as well as at an intermediate developmental temperature (22°C). However, we did not generate enough flies at 18°C, and so we only have data for flies raised at 22°C and 29°C. Our results demonstrate that 22°C is a sufficiently low temperature to detect effects of both genotype and developmental temperature on courtship success (see below). All larvae from each male strain were reared in 32 oz plastic containers on the same wheat bran diet described above (*Hamm et al., 2009*). Upon emergence, unmated male and female progeny were separated and fed water and a 1:1 mixture of dry milk:sugar ad libitum until assays were conducted. Assays of males raised at 22°C were conducted 10–11 days after eclosion, while those of males raised at 29°C were conducted 6–7 days after eclosion. This ensures that males were assayed at similar physiological ages. Females were all raised at 25°C, and unmated females were aged 8–9 days after eclosion.

We followed a similar protocol as in a previous experiment testing copulation latency in *D. melanogaster* (*Shorter et al., 2016*). Briefly, five males from a single strain were aspirated without anesthesia into an 8 oz container covered with a fine mesh cloth secured by rubber band. Five LPR females were similarly transferred into the container, marking the start of the courtship assay. The house flies were then observed every 10 min over the course of 4 hr. Copulation latency was determined in two ways. First, we measured the amount of time elapsed between the start of an assay and each mating within a container, defined as a male remaining attached to a female for at least 1 min (*Hamm et al., 2009*). Male house flies typically remain attached to females for >60 min (*Bryant, 1980*), making it unlikely, although possible, for us to miss matings within 10 min intervals. Individuals who did not mate were excluded from this analysis. Second, we used a binary variable noting whether each male mated during the 4 hr assay. Although we were unable to distinguish between individual males in this assay, we did not observe any males mate more than once within 4 hr in a pilot study conducted between one male and five females, suggesting that observed matings were by different males. All trials were conducted at 22–23°C.

To determine the effects of male type on the amount of time taken to mate, we used the glmer() function in the lme4 package in R (*Bates et al., 2015*) to create a mixed effects model as follows:

$$L \sim G + T + G \times T + b + s,$$

including male genotype (G), developmental temperature (T), and their interaction as fixed effects, batch (b) and strain (s) as random effects, and our response variable as copulation latency (L) in minutes. For the binary measure of copulation latency, we used a binomial logistic regression of the same model, with whether a fly mated as our dependent variable. We then assessed significance of fixed effects (type II sum of squares) using the Anova() function in the car package in R (*Fox et al., 2013*). Pairwise comparisons between male types (III$^M$, Y$^M$, and LPR) were conducted using Z-tests of proportions.

Lastly, we conducted a survival analysis, with unmated males treated as right-censored observations since information about their mating latency was incomplete. Specifically, we used the Surv() function in the coxme package in R (*Therneau, 2022*) to convert our copulation latency measurements (in minutes) to a right-censored survival object. We then created a Cox proportional hazards regression model using the same parameters as above, with copulation latency (in minutes) as our response variable. We assessed significance of fixed effects using analysis of deviance for a Cox model using the Anova() function, and we report hazard ratios (HRs) and their 95% confidence intervals for copulation latency effect sizes (*Burke and Holwell, 2021*).

## Assaying copulation latency in CG2120 upregulated *D. melanogaster*

We used CRISPR/dCas9 transcriptional activation (CRISPRa) to increase the expression of *CG2120* in *D. melanogaster*. We used a strain that expresses single guide RNA (sgRNA) targeting *CG2120* (BDSC ID 79962), from the Transgenic RNAi Project CRISPR Overexpression (TRiP-OE) VPR collection (*Ewen-Campen et al., 2017*). We crossed males with the sgRNA to females carrying a transgene encoding a deactivated Cas9 protein (dCas9) expressed under one of two Gal4 drivers (Gal4>dCas9). One Gal4>dCas9 strain expresses Gal4 under regulation of the *apolpp* promoter (BDSC ID 67043), which expresses in the fat body (*Brankatschk and Eaton, 2010*; *Van De Bor et al., 2015*). The other Gal4>dCas9 strain regulates Gal4 under the *elav* promoter (BDSC ID 67038), which expresses in neurons (*Sink et al., 2001*; *Zhang et al., 2002*). We collected male progeny from the cross between the sgRNA strain and each of the Gal4>dCas9 strains to assay copulation latency. Control males were generated by substituting the *CG2120* sgRNA strain with a strain carrying an sgRNA that targets the QUAS sequence from *Neurospora crassa* (BDSC stock 67539). All flies were raised at 25°C on a medium consisting of cornmeal, yeast, sugar, and agar.

We measured copulation latency in *CG2120* upregulated males and control males. In each experiment, we used a single unmated male from one of the crosses described above. That male was combined with a single unmated female from either the CantonS or OregonR strain in a vial containing our standard medium. The flies were observed for up to 2.5 hr at 25°C, and the time at which they began to copulate was recorded as our measurement of copulation latency. Fly pairs that did not copulate were excluded from the analysis. Experiments were performed in four batches for the *apolpp* driver and five batches for the *elav* driver.

We compared the fit of linear models in order to test for differences in copulation latency between *CG2120* upregulated males and control males. We first constructed a model that captured all variables in our experiment,

$$C \sim M + F + M \times F + b,$$

which included the fixed effects of male genotype (M, which can be control or *CG2120* upregulated), female strain (F, for CantonS or OregonR), and the interaction between male genotype and female strain, as well as the random effect of experimental block (b). We used the lmer() function in the lme4 package in R (*Bates et al., 2015*) to determine how each variable affects copulation latency (C). We compared that full model with one excluding the interaction term with a $\chi^2$ test in the Anova() function in R (*R Development Core Team, 2019*) in order to test if including the interaction term offers a significantly better fit. If the interaction term did not significantly improve the fit of the model, we compared the model without the interaction term with one without any effect of the female strain (i.e. F excluded from the model). If there was not a significant difference in fit between the models with

and without F, we modeled female strain as a random effect. To test for an effect of *CG2120* upregulation on copulation latency, we compared the fit of the model with male genotype as the only fixed effect and two random effects (f and b) with a model that only had the random effects (i.e. no fixed effects) using the Anova() function.

## Acknowledgements

Rebecca Presley assisted with house fly courtship assays, and Poala Najera assisted with *Drosophila* courtship assays. This work was completed in part through the use of the Maxwell Cluster provided by the Research Computing Data Core at the University of Houston. This material is based upon work supported by the National Science Foundation under Grant No. DEB-1845686. Any opinions, findings, and conclusions or recommendations expressed in this material are those of the authors and do not necessarily reflect the views of the National Science Foundation.

## Additional information

### Competing interests

Kiran Adhikari: Kiran Adhikari is affiliated with Oxford Biomedica Solutions LLC. The author has no financial interests to declare. The other authors declare that no competing interests exist.

### Funding

| Funder | Grant reference number | Author |
| --- | --- | --- |
| National Science Foundation | DEB-1845686 | Richard Meisel |

The funders had no role in study design, data collection and interpretation, or the decision to submit the work for publication.

### Author contributions

Pablo J Delclos, Conceptualization, Data curation, Formal analysis, Investigation, Visualization, Methodology, Writing – original draft, Writing – review and editing; Kiran Adhikari, Formal analysis, Methodology, Writing – review and editing; Alexander B Mai, Oluwatomi Hassan, Alexander A Oderhowho, Vyshnika Sriskantharajah, Tammie Trinh, Methodology; Richard Meisel, Conceptualization, Resources, Data curation, Formal analysis, Supervision, Funding acquisition, Visualization, Methodology, Writing – original draft, Writing – review and editing

### Author ORCIDs

Pablo J Delclos ⓘ https://orcid.org/0000-0002-6186-2208
Kiran Adhikari ⓘ https://orcid.org/0000-0002-5444-7049
Richard Meisel ⓘ https://orcid.org/0000-0002-7362-9307

### Decision letter and Author response

Decision letter https://doi.org/10.7554/eLife.90349.sa1
Author response https://doi.org/10.7554/eLife.90349.sa2

## Additional files

### Supplementary files

- Supplementary file 1. Supplementary tables (1A–1H) described throughout main text.
- MDAR checklist

### Data availability

Source data and relevant code used for data analysis have been deposited in the Texas Data Repository (https://doi.org/10.18738/T8/S6PEPH).

The following dataset was generated:

| Author(s) | Year | Dataset title | Dataset URL | Database and Identifier |
|---|---|---|---|---|
| Delclos PJ, Adhikari K, Meisel RP | 2021 | A conserved trans regulatory loop involving an odorant binding protein controls male mating behavior in flies | https://doi.org/10.18738/T8/S6PEPH | Texas Data Repository, 10.18738/T8/S6PEPH |

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
