## [Editor Report]

This valuable study investigated the effect of different levels of expression of odorant proteins from the Obp56h family on behaviour depending on the locus encoding the protein, either on the sex proto-chromosomes III or Y of the house fly. The set of evidence obtained by combining observations in house fly and functional test in the model fly *Drosophila melanogaster* is solid, but the causal effect of the odorant protein expression and the fly behaviour will need to be functionally validated in house fly as well. With the causal link established directly, this paper would be of outstanding interest to evolutionary biologists and geneticists working on the dynamics of sexual selection.

---

## [Decision Letter]

**Decision letter after peer review:**

Your paper 'A conserved trans regulatory loop involving an odorant binding protein controls male mating behavior in flies' was evaluated by two reviewers and myself. I read the manuscript, the reviews and the response to the reviews. I concur with the reviewers that the work is important. The two reviewers saw the merit of the work which is very data-rich. The authors' work leveraged that two Y chromosomes (Y-M and III-M) are involved in male mating choice and compiled analyses of gene expression in M. domestica male heads to identify genes that are differentially expressed between carriers of the two Y-chromosomes, with an emphasis on odorant binding proteins. The results suggest that Obp56h is differentially expressed between the two genotypes. This gene has been implicated with variation in male mating behavior in *D. melanogaster*. The rest of the piece works with the premise that Obp56h is the responsible allele for variation in Musca and studies (i) the genetic interactions between Obp56h and other genes in the gene network and (ii) between interaction between the allele and environment (i.e., temperature). There is a tremendous amount of work in this piece and your response to reviewers does a bona fide effort to improve the piece incorporating the extensive comments from the reviewers. Nonetheless, reviewer #2 points out to an important limitation: most of the work was carried out in *D. melanogaster* and since the genetic tools for Musca are in their infancy, the inference is limited. The piece is open about this caveat. After my own reading, however, I feel this is still a major limitation of the piece and one that can't simply be addressed at the time. Since the biological system is non-model, the work is difficult and necessarily less advanced than in *Drosophila*. On the other hand, the authors make an argument for a genetic network being conserved over 50 million years and the current data collection (as impressive as it is), do not allow the authors to make such a strong statement (for reasons very clearly delineated by Reviewer #2). My take is that, as it stands, the piece will not be able to conclusively demonstrate that Obp56h is involved in male mating choice variation in Musca and that there is a trans regulatory loop that is common across 50 million years of dipteran evolution until you can do transgenics in Musca. This is a very high mark that I understand is unfeasible at the moment.

I think the work could have the potential to make the journal impact bar if the authors are able to reframe the work to better represent the strengths of the manuscript, namely the work in *D. melanogaster*. This will also mean that the claims on conservation would have to be toned down but such a revised version would better represent the findings. I understand this would be a major overhaul of the piece and even then, we would have to reassess the fit to the journal.

[Editors' note: further revisions were suggested prior to acceptance, as described below.]

Essential revisions (for the authors):

1) The reviewers would like you to be more explicit about the fact that the results obtained in *Drosophila* suggest a possible mechanism in Musca. Since reproductive systems are rapidly evolving, the conservation of certain biological phenomena cannot be simply assumed.

2) The extent of the similarity of chromosome V between strains should be made explicit.

3) A survival analysis that would include random effects would significantly strengthen the paper.

*Reviewer #2 (Recommendations for the authors):*

Different Y chromosomes segregating as stable polymorphisms in natural populations of houseflies are associated with different behavioral and other effects. The authors report that odorant binding proteins of the Obp56h family are differentially expressed between flies carrying the IIIM Y chromosome and the YM Y chromosome. They suggest that these family members mediate the courtship differences. Since Obp56h is not located on either Y chromosome, the Y chromosomes' effects on Obp56h expression must be in trans. The authors used network analysis to identify potential IIIM or YM regulators of Obp56h. They show that upregulating neural expression of a transcription factor homologous to one of those genes, and that binds up- and down-stream of Obp56h, affects copulation latency in *Drosophila melanogaster*. They hypothesize that its IIIM linked Musca domestica homolog is upregulated, causing the courtship differences in males carrying this chromosome. The analyses are appropriate and done well. The authors' models and speculations are consistent with the data but are based on correlations or cross-genera observations.

The authors addressed some of the issues raised in my review by weakening the strength of interpretations they made from correlative data. This is an improvement, but there is still concern about speculative and correlative conclusions.

The rebuttal described experimental constraints of the system, such as rearing temperature, that made it difficult or impossible to address some issues I raised. These points are well taken. Can they be made explicit in the paper's text? For example please add to the paper that "chromosome V is (nearly) genetically identical" between the strains.

I did not understand why doing qPCR for Obp56h in dissected tissues is not feasible (rebuttal to comment 3) There are concerns about using *D. melanogaster* databases as a proxy given the evolutionary distance between the genera and the Obp56h duplications in Musca.

The use of *Drosophila* to test their hypotheses was clever given the technical limitations with Musca. But its relevance to the Musca findings is not fully convincing, as the process and roles of players within it may not be conserved.

If the results of the CHC experiment support the authors' model it would strengthen the paper to add them.

---

## [Author Response]

Your paper 'A conserved trans regulatory loop involving an odorant binding protein controls male mating behavior in flies' was evaluated by two reviewers and myself. I read the manuscript, the reviews and the response to the reviews. I concur with the reviewers that the work is important. The two reviewers saw the merit of the work which is very data-rich. The authors' work leveraged that two Y chromosomes (Y-M and III-M) are involved in male mating choice and compiled analyses of gene expression in M. domestica male heads to identify genes that are differentially expressed between carriers of the two Y-chromosomes, with an emphasis on odorant binding proteins. The results suggest that Obp56h is differentially expressed between the two genotypes. This gene has been implicated with variation in male mating behavior in D. melanogaster. The rest of the piece works with the premise that Obp56h is the responsible allele for variation in Musca and studies (i) the genetic interactions between Obp56h and other genes in the gene network and (ii) between interaction between the allele and environment (i.e., temperature). There is a tremendous amount of work in this piece and your response to reviewers does a bona fide effort to improve the piece incorporating the extensive comments from the reviewers. Nonetheless, reviewer #2 points out to an important limitation: most of the work was carried out in D. melanogaster and since the genetic tools for Musca are in their infancy, the inference is limited. The piece is open about this caveat. After my own reading, however, I feel this is still a major limitation of the piece and one that can't simply be addressed at the time. Since the biological system is non-model, the work is difficult and necessarily less advanced than in Drosophila. On the other hand, the authors make an argument for a genetic network being conserved over 50 million years and the current data collection (as impressive as it is), do not allow the authors to make such a strong statement (for reasons very clearly delineated by Reviewer #2). My take is that, as it stands, the piece will not be able to conclusively demonstrate that Obp56h is involved in male mating choice variation in Musca and that there is a trans regulatory loop that is common across 50 million years of dipteran evolution until you can do transgenics in Musca. This is a very high mark that I understand is unfeasible at the moment.I think the work could have the potential to make the journal impact bar if the authors are able to reframe the work to better represent the strengths of the manuscript, namely the work in D. melanogaster. This will also mean that the claims on conservation would have to be toned down but such a revised version would better represent the findings. I understand this would be a major overhaul of the piece and even then, we would have to reassess the fit to the journal.

We thank the Reviewing Editor of *eLife* for their constructive comments about our submitted manuscript. Based on the comments raised by the editor and reviewers from Review Commons (see below), we have added several components to this study that strengthen our hypothesis that the regulatory mechanism that influence male mating behavior (copulation latency) may be conserved between *Musca domestica* and *Drosophila melanogaster*. In addition, we have made significant edits to the tone of the manuscript to highlight the strengths of our manuscript. Specifically, we reframe the main theme of this study by focusing on the fitness effects Y chromosomes can manifest through *trans* effects on autosomal genes. We further highlight these points with the addition of a functional experiment in *D. melanogaster* that verifies the role of one candidate *trans* regulator of male copulation latency (which we propose the name *speed date, spdt*). We believe that the revisions made since first submitting this manuscript to *eLife* have significantly improved the overall quality of our work, and we thank the reviewers and Reviewing Editor again for their time and consideration.

General Statements [optional]

We thank the reviewers for their positive and constructive comments about our research topic, experimental design, and analysis. The majority of revisions we were asked to make were minor, and we were able to incorporate most of the reviewer comments as edits to the manuscript. These changes have improved the quality and clarity of this manuscript. Most of the remaining comments involved experiments that are simply not feasible in the house fly system because of limited genetic tools. When experiments such as those were suggested, we toned down our conclusions in order to not overstate the interpretations of our results.

While reviewer #1 suggested mostly minor revisions, reviewer #2 raises valid concerns regarding the conclusions we can draw based on the correlational evidence we present. We found these comments incredibly helpful and believe that incorporating these comments has provided a more appropriate tone for our discussion and conclusion components of this manuscript.

Reviewer #2 also suggested some additional experiments that would improve the overall quality of this study. While we agree that these experiments (spatial expression patterns, functional genetic data from house flies, and cuticular hydrocarbon and desiccation resistance assays between III^M^ and Y^M^ house flies) would further our understanding of the functional role of *Obp56h* genes in house fly, we are unfortunately limited in the resources available to conduct some of these experiments in house fly. However, we have noted a cuticular hydrocarbon experiment we are currently conducting and analyzing as part of another project in the III^M^/Y^M^ house fly system. If reviewers believe that this data set would be a necessary component for this manuscript, we would certainly consider incorporating those data if logistically feasible within the requested period of time. In summary, we find that the revisions we have incorporated and plan to incorporate will improve the overall quality and clarity of this manuscript

Point-by-point description of the revisions

We have either incorporated all revisions, or we chose not to carry them out because they were not feasible in the house fly system. Nonetheless, we believe that our edits to the manuscript text sufficiently adjust the tone of our conclusions to reflect the experimental limitations imposed by the organismal system. Many of the revisions requested by our reviewers were minor and have been incorporated in the most recent version of the manuscript. Please find point-by-point descriptions of our revisions and responses to reviewers below, with reviewer comments in bold and our responses in plain font. Note that line numbers that are mentioned in our responses are referencing the line numbers in the clean version of our manuscript, not the Track Changes document:

Reviewer #1:I found the description of the statistical analyses of the phenotypic assays to be a bit hard to follow – it's not 100% clear to me from reading the text how these models were set-up, and no statistical tables are included that make the model formulation more obvious. I would suggest adding more some more information of this type to the supplementary information (e.g. model scripts and tables with model outputs). I don't have any particular concerns that the statistical analysis is incorrect, and there are no issues when it comes to replication, but since the phenotypic data was analysed in several different ways, it would help the clarity of the results to include this information.

We agree that the description of these analyses should be clarified. As such, we have incorporated clearer model descriptions in the Methods section of the main text (lines 336-447) and have provided model output *XX*^2^ statistics where appropriate within the Results section (lines 493-561). We believe that these edits help provide greater clarity to these methods and results.

I think the main question could be stated more explicitly in the abstract.

We have tried to state our research objective more clearly in the abstract by stressing the goal to identify the genetic basis of mating differences observed between III^M^-Y^M^ proto-Y chromosome genotypes in nature (lines 17-27).

Some more information about the Y^M^ /III^M^ system is probably needed in the introduction for the general reader.

We have provided a more detailed explanation in the introduction and include citations that describe this system in greater detail (lines 73-92).

Lines 213-216: How many is this out of the total number of DE genes in the *Drosophila* data? I.e how representative are these genes?

A total of 50 *D. melanogaster* genes were DE upon *Obp56h*-knockdown. We were able to identify 20 *M. domestica* transcripts that are orthologous to these genes, representing 40% of the gene list. We have added this information to line 218 in the main text.

Line 368-370: This suggests that the III^M^ genotype should be more successful at higher temperatures. It could be good to state this explicitly, and remind the reader whether this prediction is consistent with the pattern seen in natural populations.

We have added a sentence to lines 493-497 in the main text clarifying this point. Specifically, we add the prediction that, if *Obp56h* expression predicts mating success in house fly as it does in *D. melanogaster*, then, at least at warm developmental temperatures, III^M^ males should outcompete Y^M^ males for access to female mates.

Line 404-405: Although it's good to mention this caveat, I don't think it's a very likely explanation. Results in *Drosophila* suggest that rearing temperature affects mating success more than courtship temperature.

This is a good point, and we have removed this point for clarity.

Lines 419-427: It could also be worth carrying out a survival analysis on this dataset, in order to be able to include data from the males that didn't mate (as an alternative to the binomial analysis of mating success). I don't think this is likely to change the conclusions, though.

We like this idea and have included this analysis in our manuscript (Methods on lines 398-407; Results on lines 527-536; Figure S2). As the reviewer anticipated, the conclusions did not change, but we think this analysis provides a clear depiction of III^M^ male house flies mating more quickly, on average, than Y^M^ house flies at 22°C.

Figure 6B: It looks like this data would be more appropriately analysed using a t-test rather than a regression, since the two treatments differ so much in expression. I doubt this will affect the outcome, though.

This is a good point, and understandable considering the low variation in *Obp56h* expression within *D. melanogaster* treatments (*Obp56h*-knockdown vs control). The reviewer is correct that the *t*-test analysis does not affect the significance of the outcome (*p* < 0.05). We first identified this pattern between *Obp56h* and *CG8745* expression from the co-expression network data in house fly. and we ultimately depicted the *Drosophila* expression data (Figure 6B) as a regression to more simply portray the analogous relationship in expression seen across the two species.

However, we have included the *t*-test results for this figure in lines 737-740.

Line 760: This recent paper might also be relevant to cite here:https://www.pnas.org/content/118/8/e2003359118

The section that the reviewer was originally referring to was removed in order to improve the clarity of the discussion.

The analytics are done well, and the results are consistent with the authors' model. However, the findings are all correlational, making the larger conclusions too speculative and uncertain. The overly speculative conclusions include elimination of temperature as a contributor to maintenance of variation and the model that Obp expression explains the differences in copulation latency and is part of a conserved trans-regulatory loop. Without functional genetic data for Obp56h in housefly, much of the language in the discussion should be tamped down to better match the strength of the evidence.

We understand that this was one of the more substantial concerns that this reviewer had with the manuscript, and we certainly don’t want to extrapolate our conclusions beyond what the correlational data show. We also note that it is not feasible to perform functional genetic experiments in house fly because we lack the molecular genetic toolkits available in *D. melanogaster* (e.g., GAL4-UAS RNAi, female germline CRISPR). As such, we have tried to incorporate cautionary statements within the discussion and conclusions sections of this manuscript to tamp down the language to better match the strength of the evidence we provide.

In addition, we have added several experiments and analyses since these reviewer comments. One such experiment was the copulation assay experiment testing the functional effect of *CG2120* in *D. melanogaster* via CRISPRa activation. We believe that these edits have helped set a more appropriate tone for our manuscript, especially relating to the discussion and conclusions. Furthermore, we believe that the addition of this experiment has provided some of the functional genetic data that the reviewer was encouraging us to include.

The authors depend heavily on functional experiments done in *D. melanogaster* to support their inferences about the likely function of Obp56h in housefly. However, the Obp56h gene family has duplicated, and presumably diverged, in the housefly lineage. Thus, the manuscript would benefit from more detailed discussion/overview on the other known roles of Obps in *Drosophila*, as well as other insects. For instance, discussion of the Obp LUSH's role in binding cVA or Obp59a's role in sensing humidity could lend further support to the author's predictions.

Obps are an extremely diverse group of genes with a wide variety of functions. While our manuscript only deals with a single Obp family (with only 1 gene copy in *D. melanogaster*), we agree with the reviewer that it would be of some value to describe the diversity of Obp functions that have been characterized in *Drosophila* and other insects. We believe that this could be especially relevant to our work because some Obps have been shown to affect multiple organismal phenotypes, which are candidate systems in which pleiotropic constraints could limit the response to selection. We now describe some of the diversity of Obps, along with the relevance to our work, within the Results and Discussion (lines 918-926).

The study would benefit from additional data on the spatial expression patterns of theObp56h family across the major chemosensory organs in head tissue (the labellum, maxillary palps, antennae) using qPCR or RT-PCR. This would provide evidence that Obp56h is expressed in these chemosensory tissues in the housefly and may clarify relationships with potential interacting genes, like the gustatory receptor.

We agree that these assays would certainly add clearer evidence towards the functional relevance of some of the correlations in gene expression we observed (in particular the relationship between *Obp56h* genes and *Gr98c*). Unfortunately due to logistical reasons, this is currently unfeasible for us to carry out within a reasonable time in house fly. However, we have mined available microarray and RNA-seq from *D. melanogaster* to identify where in the head *Obp56h, Gr98c*, and other hub genes are expressed. Using the FlyAtlas microarray dataset, we confirmed that *Obp56h* is highly expressed in head. *Obp56h* is also moderately expressed in the brain, but not in the eye. In contrast, *Gr98c* is expressed very low in all head tissues. We also looked at the expression of other hub genes in the co-expression network. The major locus of co-expression for *Obp56h* and the hub genes appears to be in the labellum. We now report this in the revised manuscript (Methods on lines 226-251; Results on lines 746-751, and Figure S7).

Additionally, the authors argue that Obp56h expression may be pleiotropically constrained due to tradeoffs with stress response and highlight desiccation resistance in particular as a potential mechanism. The study would benefit from experiments that test this hypothesis directly by comparing desiccation resistance among the two male genotypes. Though a potential caveat of this experiment is that it's not clear whether differences in Obp56h expression would affect CHC level as it may function for chemo-sensation, not production, either way the results would provide useful information.

We agree with the reviewer’s comment here. We are actually currently conducting a series of cuticular hydrocarbon (CHC) assays in this house fly system for a separate research project. While the experiment remains to be completed and analyzed, we would perhaps be willing to incorporate these data if reviewers find that this CHC data set would be a necessary component of this current manuscript.

Figure 2A, B need the 100 mark on their Y-axes. The sample size in panel B is small. Is each of its points the average of the 10 replicate samples? How consistent were all 10 replicates' results?

We have added the 100 mark on the Y-axes of Figures 2A and 2B, making it consistent with the Y-axis of panel C. The reviewer is right that each point in panel B is a measure based on one experimental batch (i.e., each data point is a batch). There are 5 batches for III^M^ males and 5 batches for Y^M^ males. Each batch consists of 10 competitive mating assays. Each competitive assay in Figure 2B consists of combining one female with one 18°C male and one 29°C male. Therefore, a single data point in this panel represents the proportion of 18°C males that successfully mated relative to the number of successful matings that occurred across the 10 assays in a batch. The consistency in this measure across batches (i.e., % 18°C males that successfully mated) can be seen in panel B. The reviewer should note that, if temperature-dependent *Obp56h* expression were highly predictive of male mating success, we would expect that III^M^ males raised at 29°C to outperform III^M^ males at 18°C, which is clearly not the case even with only 5 experimental batches of III^M^ males. We have edited the discussion of these results in lines 500-506.

Moreover, we would like to point out to the reviewer that these assays involving males raised at two different temperatures are especially challenging to perform because it is difficult to obtain males of the same age that were raised at two different temperatures. This is because temperature greatly affects developmental rate, and syncing development at 2 temperatures is hard to do. We only performed assays when we were able to adequately sync the development of flies at 18°C and 29°C.

Relatedly, Figure 2AandB are confusing to interpret without being able to see all of the datapoints (for instance, the text says "IIIM males were more successful than YM males regardless of developmental temperature" but the graph only shows the percentage of IIIM males that had mated, presumably they did better than YM males because the median is above 50%?). It may help to plot the data in a different way (stacked bar chart, or maybe total % of males mated with the genotypes in 2 different colors so that any difference is more apparent).

Figures 2A and B shows the % of III^M^ males mated in an assay where the outcome could either be III^M^ or Y^M^. Since this is a dichotomous mating assay, the percent of Y^M^ males that mated in these assays (or the percent of 29°C that mated in the case of panel B) is simply the remaining percentage. This could be ambiguous in a mate choice assay conducted with >2 male types, but it is not ambiguous in our work. Each data point shows the results from one experimental batch, with the average across batches shown as a horizontal line. If reviewers believe that presenting this as a stacked bar chart is more informative, we would be willing to make this change, but the batch level data would be missing if we only summarized across batches.

It is problematic to interpret tests of mating latency that were done at the same temperature, for animals raised at different temperatures.

We agree that this result is limited in that sense, and we would ideally like to perform all mating assays at multiple temperatures. However, there is a limited temperature range within which mating assays can be performed in house fly. At temperatures below 25°C, matings do not occur frequently enough within a reasonable time frame for us to gather sufficient experimental data. However, as reviewer 1 pointed out, developmental temperature might be more important than courtship temperature.

Line 423, 432, having few successful matings makes it impossible to measure copulation latency

This comment is referring to the result that many males, particularly those developed at 29°C, did not mate within the 4 hours of the behavioral observations. We agree that greater sample sizes, or perhaps a longer observation period (> 4 hrs) is needed to more accurately measure copulation latency for the two lines reviewer 2 refers to. For that reason, we have tried to focus more on the more robust behavioral results obtained from the single-choice mating assays (such as the significantly greater proportion of III^M^ males that mated within 4 hrs when developed at 22°C, as shown in Figure 2C). Despite the limitations of measuring copulation latency in minutes or the fact that flies reared at 29°C mated infrequently, we felt that omitting these results entirely would confuse readers more than leaving them in the text.

Chromosome V could have diverged in the different populations. The assertion that itis unlikely to differ between YM and IIIM lines is not supported by data.

We worked with nearly isogenic strains in which either the Y^M^ or III^M^ chromosome was placed on a common genetic background. We are very confident that the Chromosome V is (nearly) genetically identical across most of the strains. We know the genetic ancestry of these strains because we created them by backcrossing. They are all inbred for the same alleles on that common genetic background. While there may be some new mutations that occurred in the strains subsequent to their creation, those mutations are unlikely to have meaningful effects for two reasons. First, they would have to be generated by new mutations (i.e., not previously existing segregating variation) given the shared inbred genetic background of all strains. Second, the time since creation is so short that it is highly unlikely that those new mutations rose from a frequency of 1/2N to (near) fixation.

We did include some strains with different backgrounds, but we were careful to consider the background effect in those data. For example, we repeated the RNA-seq analysis by excluding those strains with different backgrounds and obtained similar results (see lines 570-578). Second, we performed some of our assays using males from the LPR strain (see Figure 2C), which has a different background from all other strains used in the mating assays. This was done deliberately to determine the effect of the LPR background on copulation latency because the females come from the LPR strain. Importantly, we find that the III^M^ chromosome on the common genetic background reduces copulation latency beyond the effect of the LPR background.

For these reasons, we are quite confident that chromosome V is NOT the cause of the phenotypic differences between the Y^M^ and III^M^ strains we used in our experiments. This is a core component of our inference that *trans* effects of III^M^ (and/or Y^M^) are responsible for expression differences of *Obp56h* genes.

Page 18, It is hard to know how meaningful cross-genera comparisons are particularly when the directions of the effects differ between the two genera.

There are two inferred directions of effect. First, genes that are downregulated upon knockdown of *Obp56h* in *D. melanogaster* have house fly orthologs that are more downregulated in III^M^ male house flies than expected by chance (Figure S6A in the revised manuscript). This cross-genera comparison is in the same direction in both species. We think that this is notable and worth emphasizing in the manuscript. Second, genes that are upregulated upon knockdown in *D. melanogaster* have house fly orthologs that tend to be downregulated in III^M^ house flies (Figure S6B). We think that the reviewer is referring to this result, which is in opposite directions. Therefore, one of two cross-genera comparisons have effects in the same direction. As for how meaningful cross-genera comparisons are, it is certainly true that these are not definitive proofs of evolutionary conservation, but we find the result promising enough to warrant future investigation. We have elaborated on this specific result in lines 648-661.

Several other conclusions are too speculative, such as whether Md-gd participates inToll signaling, or the basis for suggesting LOC1010893651 as the transcription factor regulating Obp56h.

We have taken the reviewer’s general concern regarding the over-emphasis of our results into consideration during our revisions. We believe that the changes made throughout the results and discussion have emphasized the correlative nature of this study as well as the need for functional experiments to more confidently prove the conclusions we have drawn. Nonetheless, there is substantial evidence in *D. melanogaster* that *gd* encodes a serine protease involved in activation of Toll signaling (LeMosy et al. PNAS April 24, 2001 98 (9) 5055-5060; https://doi.org/10.1073/pnas.081026598). This function is likely conserved across flies given the conservation of the protein coding sequence (60% amino acid identity between the protein coding sequences, including both the Trypsin-like serine protease region and the canonical Nterminal region). Similarly, we speculate that *LOC1010893651* may be the transcription factor that regulates *Obp56h*, and we emphasize the “may” in the revisions. In addition, we believe that our additional experiment testing the effect of CRISPRa activation of *CG2120* in *D. melanogaster* adds even further evidence for the relationship between *CG2120* and reduced copulation latency in flies with reduced expression of *Obp56h*.

Line 522: "…DE *D. melanogaster* X chromosome genes Obp56h knockdown flies" should have "in" between genes and Obp56h.

We have corrected this typo.

Lines 556-558: it seems a bit too speculative to hypothesize the trans regulatory feedback loop between house and fruit flies works through similar molecular functions based on a GO term, especially when that GO term is as vague as "stress response"

We agree that basing the argument on a single GO term, especially one as vague as “stress response”, is not a solid foundation for similar molecular functions underlying the *trans* regulatory loop in house flies and *Drosophila*. What we intended to say was that the shared GO term contributes to the other evidence we report for an evolutionarily conserved *trans* regulatory loop. We believe that the results described elsewhere in the paragraph (summarized in Figure S5) and results shown later (e.g., Figures 4, 6, and 8) are much stronger evidence for a conserved molecular pathway, whereas the GO term similarity is simply another (perhaps less substantial) piece of evidence. We have clarified this in the revisions (lines 648-661).

Line 726, the sex chromosome of Musca domestica and *Drosophila* melanogaster may have originated from the same autosome, but they are unlikely to still be the "same" chromosome. They have likely diverged between the two lineages.

We should have been clearer in the manuscript. House fly and *Drosophila* do not share the same sex chromosome by common ancestry. The chromosome was autosomal in the most recent common ancestor (MRCA) of house fly and *Drosophila*. It has independently become sex-linked in the lineage leading to *Drosphila* and in house fly. We have rephrased this to more clearly explain this convergent evolution (lines 1019-1031).

[Editors' note: further revisions were suggested prior to acceptance, as described below.]Essential revisions (for the authors):1) The reviewers would like you to be more explicit about the fact that the results obtained in *Drosophila* suggest a possible mechanism in Musca. Since reproductive systems are rapidly evolving, the conservation of certain biological phenomena cannot be simply assumed.

We believe that our manuscript edits in response to reviewers have largely addressed this concern, resulting in a manuscript that provides evidence, but does not definitively conclude, that the genetic pathway described is largely conserved.

2) The extent of the similarity of chromosome V between strains should be made explicit.

We have edited our explanation of the strains used in this experiment to more clearly describe their genetic similarities (see response to Reviewer 2 below)

3) A survival analysis that would include random effects would significantly strengthen the paper.

The modified survival analysis has been incorporated.

Reviewer #2 (Recommendations for the authors):Different Y chromosomes segregating as stable polymorphisms in natural populations of houseflies are associated with different behavioral and other effects. The authors report that odorant binding proteins of the Obp56h family are differentially expressed between flies carrying the IIIM Y chromosome and the YM Y chromosome. They suggest that these family members mediate the courtship differences. Since Obp56h is not located on either Y chromosome, the Y chromosomes' effects on Obp56h expression must be in trans. The authors used network analysis to identify potential IIIM or YM regulators of Obp56h. They show that upregulating neural expression of a transcription factor homologous to one of those genes, and that binds up- and down-stream of Obp56h, affects copulation latency in *Drosophila melanogaster*. They hypothesize that its IIIM linked Musca domestica homolog is upregulated, causing the courtship differences in males carrying this chromosome. The analyses are appropriate and done well. The authors' models and speculations are consistent with the data but are based on correlations or cross-genera observations.The authors addressed some of the issues raised in my review by weakening the strength of interpretations they made from correlative data. This is an improvement, but there is still concern about speculative and correlative conclusions.

We agree that the results from our correlative analyses should not be interpreted as causal evidence, and we believe that the changes incorporated in this round of edits have reduced the potential for readers to interpret our conclusions as overly speculative.

The rebuttal described experimental constraints of the system, such as rearing temperature, that made it difficult or impossible to address some issues I raised. These points are well taken. Can they be made explicit in the paper's text? For example please add to the paper that "chromosome V is (nearly) genetically identical" between the strains.

We have incorporated this change. Specifically, in Lines 183 – 187, we have provided more transparency by noting that these flies, although from a common genetic background, are unlikely to be entirely genetically identical across their genomes.

I did not understand why doing qPCR for Obp56h in dissected tissues is not feasible (rebuttal to comment 3) There are concerns about using *D. melanogaster* databases as a proxy given the evolutionary distance between the genera and the Obp56h duplications in Musca.

We agree that only analyzing data from *D. melanogaster* is less than ideal, and there are limitations to using these data as a proxy for house fly gene expression. We have added a statement noting that tissue-specific expression data from house fly would be needed to gain further insight on the function of this module in lines 339-340. Ideally, we would have looked at tissue-specific expression data in Musca domestica in addition to *D. melanogaster*. However, there are two considerable challenges to this. First, measuring expression of the multiple Obp56h genes in house fly using qPCR would require designing paralog-specific primers and/or probes, which we have determined is not feasible given the sequence similarity between paralogs (this is not as big of a challenge with RNA-seq data because we used a probabilistic read mapping approach that can model read mapping ambiguity). While we could use qPCR to measure expression of the candidate trans regulators in our network in house fly (e.g., CG2120), there is a second challenge to doing so: we do not have the technical expertise to perform tissue dissections to collect comparable data as available in the *Drosophila* RNA-seq datasets. Therefore, to measure Obp56h gene e expression in house fly tissues samples similar to those available in the *Drosophila* RNA-seq data we analyzed would require fine-scale tissue dissection followed by RNA-seq, which we believe is beyond the scope of the work we presented in the manuscript. We believe that the findings from the *D. melanogaster* data sets we were able to analyze, although not of primary importance, add to our overall understanding of the potential function of the Obp56h gene family and is therefore worth including.

The use of *Drosophila* to test their hypotheses was clever given the technical limitations with Musca. But its relevance to the Musca findings is not fully convincing, as the process and roles of players within it may not be conserved.

We agree that a lack of evolutionary conservation may limit some of the conclusions we draw from our data. We have attempted to be clear in our manuscript about those limitations and their implications. However, we believe that the vast amount of concordant and corroborating evidence (though often correlational) provided in this manuscript suggests that the genetic pathway that influences copulation latency has been conserved to a great extent. While we could collect additional data to add further evidence of the level of conservation between these two systems, these additional datasets would be time consuming, labor intensive, and expensive to collect (e.g., RNA-seq from house fly dissected tissue samples, discussed above). We are also concerned about adding a large data set to a manuscript that has already been noted as being data-dense. That said, we are currently pursuing follow-up experiments (RNA-seq from above, CHC profiling discussed below) that could build upon the results presented in our manuscript. Some of these data may be available to include in a revised manuscript if the reviewers require it, although this will delay publication substantially.

If the results of the CHC experiment support the authors' model it would strengthen the paper to add them.

We agree with this reviewer that the CHC experiment would indeed provide invaluable insight into the mechanism that might maintain this polymorphism of proto-Y chromosomes in nature. However, as the first reviewer noted, this is already a fairly data-dense manuscript, and we worry that incorporating additional studies at this point may come at the risk of hurting the clarity of our findings.